# NEGATIVE PRE-ACTIVATIONS DIFFERENTIATE SYNTAX

**Linghao Kong**[1], **Angelina Ning**[1], **Micah Adler**[1], **& Nir Shavit**[1,2]
[1]MIT   [2]Red Hat AI
{`linghao, angn_731, micah432, shanir`}@mit.edu

## ABSTRACT

Modern large language models increasingly use smooth activation functions such as GELU or SiLU, allowing negative pre-activations to carry both signal and gradient. Nevertheless, many neuron-level interpretability analyses have historically focused on large positive activations, often implicitly treating the negative region as less informative, a carryover from the ReLU-era. We challenge this assumption and ask whether and how negative pre-activations are leveraged by models. We address this question by studying a sparse subpopulation of Wasserstein neurons whose output distributions deviate strongly from a Gaussian baseline and that functionally differentiate similar inputs. We show that this negative region plays an active role rather than reflecting a mere gradient optimization side effect. A minimal, sign-specific intervention that zeroes only the negative pre-activations of a small set of Wasserstein neurons substantially increases perplexity and sharply degrades grammatical performance on BLiMP and TSE, whereas both random and perplexity-matched ablations of many more non-Wasserstein neurons in their negative pre-activations leave grammatical performance largely intact. Conversely, on a suite of non-grammatical benchmarks, the perplexity-matched control ablation is more damaging than the Wasserstein neuron ablation, yielding a double dissociation between syntax and other capabilities. Part-of-speech analysis localizes the excess surprisal to syntactic scaffolding tokens, layer-specific interventions show that small local degradations accumulate across depth, and training-dynamics analysis reveals that the same sign-specific ablation becomes more harmful as Wasserstein neurons emerge and stabilize. Together, these results identify negative pre-activations in a sparse subpopulation of Wasserstein neurons as an actively used substrate for syntax in smooth-activation language models.

## 1 INTRODUCTION

Prior works have successfully investigated the roles of specific neurons within large language models (LLMs), identifying units tied to concepts, run-stable behavior, and confidence regulation (Gurnee et al., 2023; 2024; Stolfo et al., 2024). With respect to grammar, researchers have isolated neurons that are language selective, causally implicated in agreement, and selective for specific syntactic phenomena (AlKhamissi et al., 2024; Mueller et al., 2022; Duan et al., 2025). A common working heuristic in such analyses is to define what a neuron "represents" as inputs that produce high positive pre-activations, an assumption originating in rectified architectures, where negative values are effectively inactive (Nair & Hinton, 2010). Although recent methods such as sparse autoencoders can, in principle, be sensitive to negative activations (Jing et al., 2025; Cunningham et al., 2023), the structure of the negative pre-activation region in smooth-activation language models remains comparatively underexplored.

This gap is particularly striking given that modern transformers predominantly employ smooth activation functions such as GELU (Hendrycks & Gimpel, 2016) and SiLU (Elfwing et al., 2018). Introduced primarily for optimization benefits, these functions provide smooth gradients near zero, mitigate "dying ReLU" issues (Lu et al., 2019), and empirically improve performance (Shazeer,

---

[1]A preliminary version of this work appeared in the 3rd Workshop on High-dimensional Learning Dynamics at ICML 2025 (Kong et al., 2025).

[2]Code available at `https://github.com/Shavit-Lab/Negative-Differentiation`.

2020). Crucially, for inputs less than zero, such functions produce both nonzero output and gradient. Thus, in principle, the negative pre-activation region is available for computation, yet it is typically treated as inert. We challenge this assumption and test whether the negative pre-activations are functionally utilized, and if so, for what purpose.

To address this question, we focus on Wasserstein neurons: a recently identified subpopulation of neurons whose pre-activation distributions exhibit large Wasserstein distance (WD) from a Gaussian baseline (Sawmya et al., 2025). Prior work has shown that such neurons, though they comprise only a small fraction of the network, are disproportionately sensitive to sparsification and targeted removal. Functionally, they uniquely map locally similar input vectors to widely separated output scalars via their dot product, a property quantified as mapping difficulty (MD). Following prior usage, such neurons are termed entangled, extending concepts from superposition in which multiple features are shared by the same neuron (Elhage et al., 2022; Adler et al., 2025) to this complementary case in which closely related inputs are separated by a single neuron. WD and MD correlate strongly, motivating the use of WD as a practical entanglement proxy (Section A.1).

We find that Wasserstein neurons in the linear projections immediately preceding the nonlinearity of the multilayer perceptron (MLP) block (the gate projection in GLU-style models (Shazeer, 2020) such as Llama (Grattafiori et al., 2024) and the up projection in GPT-2-style models (Radford et al., 2019) such as Pythia (Biderman et al., 2023)) share an interesting property: the deviation from Gaussianity concentrates in the negative region of the pre-activation space, and so we focus on this tractable subset of neurons as candidates for analysis. This effect is markedly stronger in non-ReLU models: although ReLU-based models such as OPT (Zhang et al., 2022) also exhibit non-Gaussian pre-activation distributions, theirs show comparatively less concentration of such structure specifically in the negative region, consistent with ReLU's clamping (Figure 1).

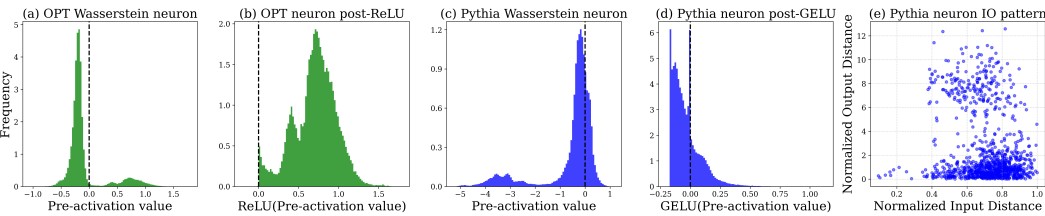

Figure 1: Wasserstein neurons in ReLU vs non-ReLU LLMs. (a, b) In OPT-1.3B, a ReLU-based model, the dominant pre-activation mass resembles a somewhat Gaussian peak whose mode lies below zero, with an additional mildly multimodal positive tail. (c, d) In Pythia 1.4B, a GELU-based model, the dominant mass instead centers near zero, and the negative pre-activation region exhibits more pronounced multimodality, reflecting preservation of negative inputs. (e) The input output (IO) relationship of the Pythia Wasserstein neuron, showing that for pairs of inputs that are fairly similar, their outputs are still mapped far apart by the neuron. More details provided in Section 2.1. Neurons acquired from the up projection in the second MLP block of their respective models.

Here, we show that this seemingly "inactive" region is in fact crucial for model function: Wasserstein neurons systematically exploit negative pre-activations to differentiate syntax. Across multiple LLM families, clamping only the negative pre-activations of the top few Wasserstein neurons in each MLP gate projection significantly impairs general model performance, yielding large perplexity increases. Matching the same perplexity rise with non-entangled neurons requires far more units. Even under these perplexity-matched conditions, only the Wasserstein-neuron intervention produces large drops in grammatical accuracy on BLiMP (Warstadt et al., 2020) and TSE (Marvin & Linzen, 2018), whereas the non-entangled neuron ablation produces larger drops on a panel of non-grammatical benchmarks, yielding a clear double dissociation. Further analysis reveals that these neurons' input differentiation preferentially separates syntactic scaffolding tokens, such as determiners and prepositions, sending locally similar inputs to distinct negative pre-activation values. These effects are strongest in early layers and compound across depth. Across training checkpoints, the same fixed negative pre-activation ablation grows increasingly damaging as Wasserstein neurons acquire their characteristic non-Gaussian structure. Together, these results indicate that the negative pre-activation region serves as an active site of computation in this entangled subpopulation and is disproportionately implicated in syntax, rather than serving as a mere optimization convenience.

## 2 A MOTIVATING CHARACTERIZATION OF WASSERSTEIN NEURONS

### 2.1 IDENTIFYING WASSERSTEIN NEURONS IN LANGUAGE MODELS

First, we specify where in LLMs we analyze Wasserstein neurons and briefly detail their previously established conventions, such as how their WD and MD metrics are calculated. The MLP block of GPT-2-style GELU-based models, such as the Pythia suite (Biderman et al., 2023), is as follows: $\boldsymbol{y} = \boldsymbol{W}_{down}(\text{GELU}(\boldsymbol{W}_{up}\boldsymbol{x}))$. The MLP of SiLU-based GLU-style models, such as Llama 3.1 8B (Grattafiori et al., 2024), Mistral 7B v0.3 (Jiang et al., 2023), and Qwen3 8B Base (Yang et al., 2025) is as follows: $\boldsymbol{y} = \boldsymbol{W}_{down}(\text{SiLU}(\boldsymbol{W}_{gate}\boldsymbol{x}) \odot (\boldsymbol{W}_{up}\boldsymbol{x}))$. Note that the naming convention for the linear projection preceding the nonlinearity is flipped. We follow the convention that, when treating $\boldsymbol{x}$ and $\boldsymbol{y}$ as column vectors, neurons are defined as row vectors in $\boldsymbol{W}$. We examine Wasserstein neurons in $\boldsymbol{W}_{up}$ in Pythia 70M to 12B as well as in $\boldsymbol{W}_{gate}$ in Llama 3.1 8B, Mistral 7B v0.3, and Qwen3 8B Base. We use Pythia to better investigate training dynamics through their publicly released training checkpoints, and we use the other three models to investigate the phenomenon in more modern language models.

To compute the WD and MD of a neuron with weights $\boldsymbol{w}$, real input text totaling $N$ tokens is fed into the model. We use the test set of WikiText 2 (Merity et al., 2016). The distribution of input vectors into a neuron, $\{\boldsymbol{x}_i\}_{i=1}^N$, as well as the distribution of the output scalars from its dot product computation, $\{y_i\}_{i=1}^N = \{\boldsymbol{w}^T\boldsymbol{x}_i\}_{i=1}^N$, are collected. An example of this output distribution is shown in Figure 1c. $\{y_i\}_{i=1}^N$ is normalized to have zero mean and unit variance, and the Wasserstein distance of this normalized distribution with a unit Gaussian is calculated as WD. Unless otherwise specified, in the following sections the WD of a neuron always refers to the Wasserstein distance of its normalized output distribution to a unit Gaussian.

To compute MD, pairs of input vectors $\boldsymbol{x}_i, \boldsymbol{x}_j$ are randomly selected and their $L^2$ norm is calculated, as well as the $L^2$ norm of their corresponding outputs $y_i, y_j$. Each input pair $L^2$ norm is normalized by the maximum of the set, and each output pair $L^2$ norm is normalized by the median of the set. An example of these pairs for a Wasserstein neuron is shown in Figure 1e. Finally, each normalized output pair difference is divided by their corresponding normalized input pair difference, and the average of these ratios is calculated as the MD of a neuron to summarize how far apart neurons map similar inputs. Because WD and MD are proxies of each other, we primarily use WD to select for entangled neurons, and we use MD when specifically targeting pairs of inputs that are mapped far apart, which will be specified in the relevant sections. We further investigate the potential confounds of the influence of asymmetry and kurtosis on WD in Section A.1.

### 2.2 EVALUATION PROTOCOLS

We evaluate syntactic behavior using two complementary benchmark suites, BLiMP (Warstadt et al., 2020) and TSE (Marvin & Linzen, 2018). BLiMP (the Benchmark of Linguistic Minimal Pairs) consists of 67 sub-datasets with 1,000 minimally different pairs of a grammatical and ungrammatical sentence. This benchmark covers 13 broad categories spanning syntax, morphology, and semantics, such as subject-verb agreement, determiner-noun agreement, and ellipsis. TSE (Targeted Syntactic Evaluation) is a fine-grained challenge set of roughly 350K minimally different sentence pairs focusing on three families of structure-sensitive dependencies: subject-verb number agreement, reflexive anaphora, and negative polarity item licensing. TSE stress-tests specific hierarchical dependencies and complements BLiMP's broader coverage. For both benchmarks, accuracy is the fraction of pairs for which the model assigns higher total log-probability to the grammatical sentence.

To assess non-grammatical performance, we utilize the Language Model Evaluation Harness (Gao et al., 2024) to test model performance on ARC Challenge, ARC Easy (Clark et al., 2018), BoolQ (Clark et al., 2019), HellaSwag (Zellers et al., 2019), PIQA (Bisk et al., 2020), SciQ (Welbl et al., 2017), TruthfulQA (Lin et al., 2022), and WinoGrande (Sakaguchi et al., 2021). These cover a broad set of abilities, such as science (ARC Challenge, ARC Easy, SciQ), commonsense reasoning (HellaSwag, PIQA, WinoGrande), reading comprehension (BoolQ), and truthfulness (TruthfulQA). Additionally, we track perplexity on the WikiText 2 validation set. For parts of speech (POS) and dependency tagging, we use the spaCy (Honnibal et al., 2020) English core web small model.

### 2.3 WASSERSTEIN NEURON EMERGENCE TRACKS GRAMMATICAL ACCURACY

To better characterize Wasserstein neurons and gain an intuition for their function, we analyze their development across model sizes and over the course of training. We use the Pythia suite of language models, specifically Pythia 70M to 12B. In the $W_{up}$ of the second MLP block of each model, we compute the WD of every neuron both at the final checkpoint and across training steps. We specifically follow the same Wasserstein neurons, those with the top $1\%$ WD as measured at the final checkpoint, across training to track their emergence.

First, we generally find that larger models tend to contain Wasserstein neurons with higher maximum and average WD (Figure 2a). Moreover, these neurons emerge very rapidly over training: their WD increases sharply within the first 25K steps, or about 50B tokens, after which the highest WD neurons are already distinguishable from the rest (Figure 2b). Examining their weights across checkpoints, we observe a complementary pattern. These neurons change more than average early in training and then undergo a period of relative consolidation (Figure 2c). We quantify this using cosine dissimilarity between successive 10K-step checkpoints, normalized by the layer mean. Together, these observations suggest that high-WD neurons specialize early and stabilize thereafter.

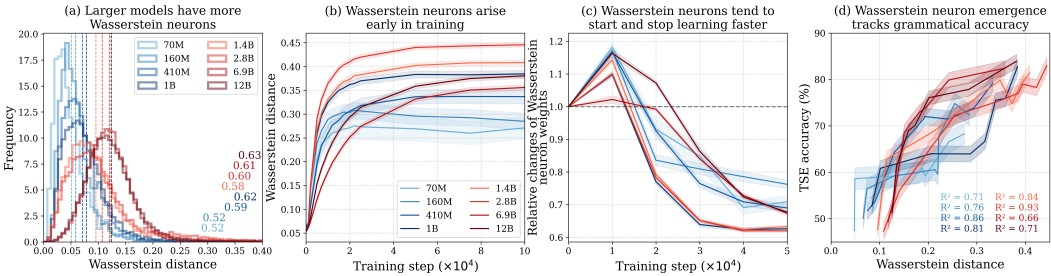

Figure 2: Wasserstein neuron emergence tracks grammatical accuracy. (a) Larger models tend to have Wasserstein neurons with greater maximum WD, and more neurons with slightly greater WD. Dotted lines indicate mean WD, and text indicates maximum WD of all neurons in layer. (b-d) share the same legend. (b) Wasserstein neurons arise rapidly during training, within roughly 50B tokens. The WD of the same cohort of Wasserstein neurons is calculated at each checkpoint. (c) Wasserstein neurons tend to start and stop learning faster than other neurons, as measured by the cosine dissimilarity, normalized to the layer average, between successive 10K-step checkpoints. (d) At various checkpoints in training, the WD of the Wasserstein neuron group is compared to the model's performance on TSE at that time, and they strongly correlate. All neurons from the up projection in each model. Shaded bands are one standard error of the mean.

Because syntactic abilities are also known to arise early in training (Duan et al., 2025; Müller-Eberstein et al., 2023), we ask whether the development of Wasserstein neurons tracks grammatical competence. Indeed, across checkpoints, the aggregate WD of a fixed cohort of high-WD neurons correlates with TSE accuracy (Figure 2d). However, this correlation alone does not determine whether these neurons are uniquely tied to syntax or simply reflect general representational capacity.

We therefore explicitly pose the question that motivates the next section: is the structure that Wasserstein neurons develop, particularly in the negative pre-activation region, causally necessary for grammatical behavior, or merely a byproduct of global improvement? In Section 3, we address this using targeted, sign-specific ablations and compare their effect on grammatical and non-grammatical benchmarks with matched-perplexity ablations.

## 3 ABLATING NEGATIVE PRE-ACTIVATIONS IN WASSERSTEIN NEURONS UNIQUELY HARMS GRAMMAR

We causally perturb Wasserstein neurons by zeroing only their negative pre-activations immediately before the nonlinearity: $a'_k = \max(a_k, 0)$ for $k \in S$, $a'_k = a_k$ otherwise, where $a$ are pre-activations in the MLP gate/up projection and $S$ contains the top $p\%$ WD neurons per layer, with $p\%$ being on the order of $1\%$ (Figure A7a). The model, weights, and nonlinearities are otherwise unchanged apart from this seemingly minor alteration.

We use two control conditions. The first perturbs an equal number of randomly selected neurons per layer using the same ablation. The second is a perplexity-matched control that perturbs the negative pre-activations of low-WD neurons. Specifically, for each layer we ablate the bottom $m\%$ of neurons as ranked by WD, where $m$ is a single global percentage applied uniformly across layers, increased until the resulting WikiText 2 perplexity matches that of the top-WD ablation. This ensures that both interventions involve comparable global degradation while differing in which neurons are perturbed.

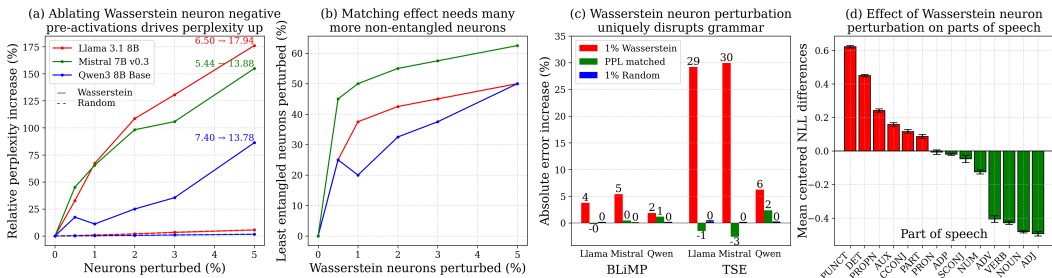

Figure 3: Sign-specific perturbation of Wasserstein neurons disproportionately harms grammar. (a, b) share the same model colors. (a) Perplexity increases when clamping only the negative pre-activations of the top-WD neurons; random controls are much smaller. Numbers indicate starting and ending absolute perplexity. (b) Matching the perplexity increase from perturbing the entangled fraction of neurons requires an order of magnitude more non-entangled units. (c) Perturbing Wasserstein neurons uniquely impacts grammatical capabilities, even compared to the perplexity-matched control. In each model, the top $1\%$ Wasserstein neurons in each layer were perturbed for the benchmark. The least entangled $40\%$ of neurons in Llama, $50\%$ in Mistral, and $20\%$ in Qwen in each layer were used as the perplexity-matched control. (d) At a per token resolution, tokens associated with syntactical scaffolding incur a much higher surprisal for the $1\%$ Wasserstein perturbation compared to the perplexity matched control in Llama 3.1 8B. NLL differences were mean shifted by the global difference. Randomly sampled controls were acquired over ten trials. Error bars indicate one standard error of the mean. Raw scores and CI's are in Table A2.

This intervention yields a striking result: although it affects only $\approx 1\%$ of neurons, and alters only their negative pre-activations, it produces disproportionately large functional damage. Perplexity increases steeply with the fraction of neurons perturbed in Llama 3.1 8B, Mistral 7B v0.3, and Qwen3 8B Base, doubling in Llama and Mistral with just a $2\%$ perturbation and in Qwen with $5\%$, far exceeding random controls (Figure 3a). Matching the same perplexity increase with low-WD neurons requires clamping vastly more units: the effect of perturbing just $1\%$ of Wasserstein neurons per layer is only matched by perturbing roughly $50\%$ of the least-entangled neurons in Mistral, $35\%$ in Llama, and $20\%$ in Qwen per layer (Figure 3b). Crucially, across all three models, only the $1\%$ Wasserstein intervention, not the random or perplexity-matched controls, yields large drops in grammatical accuracy on BLiMP and TSE, with Llama and Mistral degrading more than Qwen (Figure 3c). Token level analysis on Llama 3.1 8B localizes the added surprisal (compared to the perplexity-matched control) to syntactic scaffolding POS classes such as determiners, punctuation, auxiliaries, and particles, but not nouns, verbs, adjectives, or adverbs (Figure 3d), confirming that the negative pre-activation region of Wasserstein neurons is not inert but mechanistically necessary for syntax. Additional controls validating the importance of the sign of negative pre-activations, rather than just their magnitude, can be found in Section A.4. Repeating this analysis, but selecting neurons based on their MD instead of WD, yields the same qualitative result (Figure A3).

To test whether the observed effects are specific to syntax rather than broad capacity loss, we evaluate the same interventions on a suite of non-grammatical benchmarks. We focus on Llama 3.1 8B and reuse the three ablation conditions as before. In each MLP gate projection, we ablate the negative pre-activations of either the top $1\%$ WD neurons, $1\%$ of randomly selected neurons, or the bottom $40\%$ WD neurons. We then evaluate the interventions on eight multiple-choice benchmarks that probe general reasoning and comprehension rather than syntactic competence: ARC Challenge, ARC Easy, BoolQ, HellaSwag, PIQA, SciQ, TruthfulQA, and WinoGrande. These tasks assess scientific and commonsense reasoning, question answering, and general reading comprehension without the potential confound of open-ended text generation.

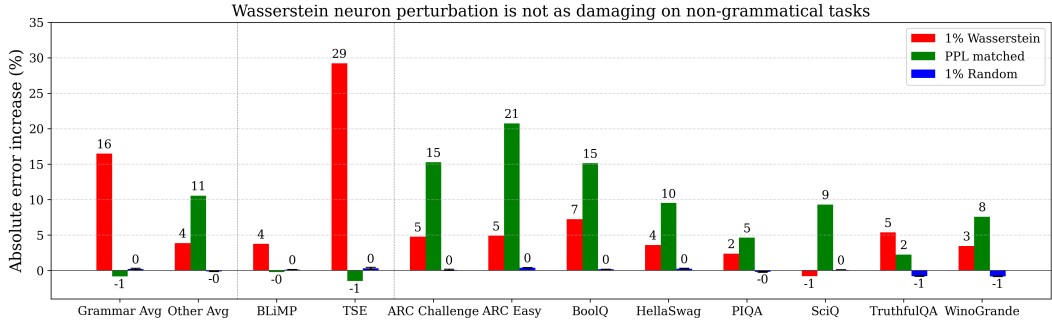

Figure 4: Non-grammatical abilities are comparatively less harmed by Wasserstein neuron perturbation. All benchmarks were run in 0-shot. Randomly sampled controls were acquired over ten trials. Error bars indicate one standard error of the mean. Raw scores and CI's are in Tables A2, A3.

The performance pattern is the opposite of what we observe for BLiMP and TSE. With the exception of TruthfulQA, the perplexity-matched ablation on the bottom $40\%$ WD neurons produces larger error increases than the top $1\%$ WD neuron ablation on every benchmark. On average across the non-grammatical tasks, the low-WD ablation yields an $\approx 11\%$ absolute increase in error relative to baseline, whereas the Wasserstein ablation yields $\approx 4\%$ and the random ablation produces no appreciable change. TruthfulQA is the only outlier, with error increasing by $5\%$ under the Wasserstein ablation compared to $2\%$ for the perplexity-matched control. Thus, when perplexity is matched, ablating many low-WD neurons primarily degrades non-syntactic capabilities, while ablating a tiny set of high-WD neurons has a comparatively milder effect on these tasks. We replicate this non-grammatical benchmark analysis on Mistral 7B v0.3 and Qwen3 8B Base and observe consistent grammar associated degradation, with some model dependent variation (Section A.3).

Taken together, these benchmarks establish a double dissociation. Clamping the negative pre-activations of just $1\%$ of Wasserstein neurons causes large drops on BLiMP and TSE but comparatively modest degradation on non-grammatical benchmarks. In contrast, clamping the negative pre-activations of many low-WD neurons leaves grammatical performance largely intact while substantially degrading general capabilities. These findings support the view that negative pre-activations in a sparse set of Wasserstein neurons play a critical role in syntactic processing, while more diffuse capacity for other tasks is distributed across the bulk of low-WD neurons, pointing to a structured organization of negative pre-activation behavior. To localize the mechanism, we move from model-level interventions to layer-level ablations in Llama 3.1 8B.

## 4 LAYERWISE ABLATIONS REVEAL EARLY LAYER ORIGINS AND CUMULATIVE ERROR BUILDUP IN SYNTACTIC STRUCTURE

To understand which layers are most crucial for particular grammatical phenomena, and to understand how damage compounds with depth, we split Llama 3.1 8B into eight groups of four successive layers. We perform group-wise negative pre-activation ablations to $1\%$ of Wasserstein neurons in each of the layers in the group. We then benchmark Llama with only a single group perturbed, or with all layers up to and including a group perturbed. Broadly speaking, across both BLiMP and TSE, early layers are much more sensitive to ablation and therefore critical for grammatical function, in line with previous works (Tenney et al., 2019; Hewitt & Manning, 2019).

Specifically for BLiMP, Wasserstein neuron perturbation sharply increases error on ellipsis and subject–verb agreement (Figure 5a), two constructions requiring non-local dependencies (Merchant, 2013; Franck et al., 2006). Binding and determiner–noun agreement are also affected, albeit to a lesser degree. Later layer groups show much weaker effects, suggesting that Wasserstein neurons in the early network establish syntactic scaffolding upon which subsequent layers depend. When ablations are applied cumulatively, error grows monotonically, especially for ellipsis, subject-verb agreement, determiner–noun agreement, and filler–gap dependencies, demonstrating that local disruptions compound across depth (Fig. 5b).

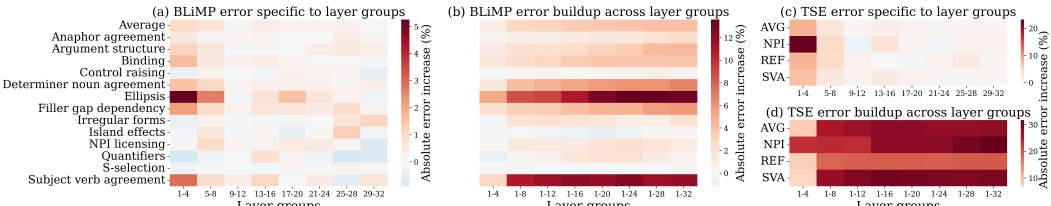

Figure 5: Individual and cumulative layerwise ablations. (a, b) share the same y-axis labels. (a) Early layer ablations yield the greatest increases in error, specifically within ellipsis and subject-verb agreement. (b) Error increases monotonically with cumulative ablation, with the strongest effects for ellipsis and subject-verb agreement. (c) TSE performance is also the most sensitive to early layer perturbation, especially for negative polarity item licensing. (d) Error for TSE grows monotonically as well. All benchmarks collected for $1\%$ Wasserstein perturbation per layer.

TSE highlights this vulnerability even more strongly. Local ablations already produce dramatic degradation, with negative polarity item licensing suffering a striking $20\%$ increase in error from just the perturbation of the first four layers (Fig. 5c). Furthermore, cumulative ablations raise error across all syntactic classes within TSE (Fig. 5d).

These layerwise effects mirror the earlier POS-level findings: disrupting negative pre-activations in Wasserstein neurons disproportionately harms the functional scaffolding of syntax (auxiliaries and determiners), and small early-layer hits compound across depth into broad grammatical failure. This analysis suggests an early, sign-specific mechanism that feeds many grammatical constraints. To make this concrete, we examine a single Wasserstein neuron in Pythia 1.4B and examine both how it separates nearby inputs and what inputs it is separating.

## 5 NEGATIVE DIFFERENTIATION OF SYNTACTIC TOKENS

Returning to the notion of Wasserstein neurons as input differentiators, we examine a particular Wasserstein neuron in Pythia 1.4B, neuron 5176, the same neuron from Figure 1c. As we feed real WikiText 2 validation data into the neuron, we sample 2,000 random tokens and form 1,000 pairs. We then calculate the ratio of the normalized output distance to the normalized input distance, as described previously (Section 2.1), and choose the pairs with the greatest ratio. We show the top ten pairs for visualization purposes in the neuron's input output relationship (Figure 6c). The most separated pairs are overwhelmingly grammatical, such as the preposition "for" being mapped far from the determiner "the" (Figure 6a). We show additional neurons in Section A.5, finding many with interpretable pairs, such as those processing coordinating conjunctions or punctuation.

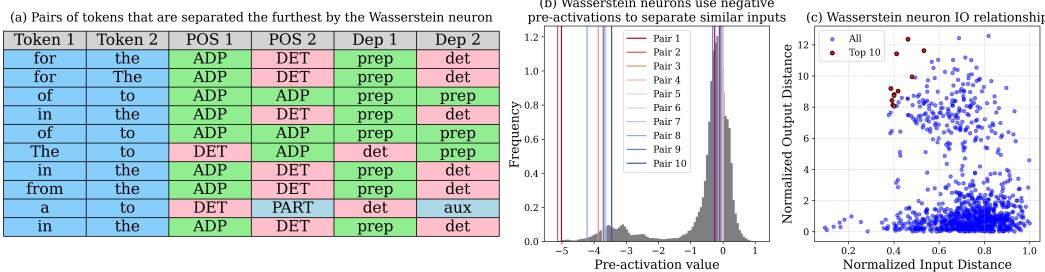

Figure 6: Representative example of Wasserstein neuron input differentiation. (a) The top ten pairs of tokens that are differentiated by Wasserstein neuron 5176, with the POS and dependency labeled. They are predominantly syntactically functional. (b) The output distribution of this neuron over WikiText 2, with the specific pair output values highlighted. Eight of the top ten differentiated pairs are driven to two very distinct negative values, rather than the perhaps more expected positive and negative value pair. (c) The top ten differentiated pairs are all fairly similar as input vectors, but are mapped very far apart by this neuron.

Counterintuitively, these most differentiated pairs are not mapped far apart because one output value is positive and the other is negative. Rather, both elements of each pair are driven negative, but to different depths (Figure 6b), accounting for eight out of the ten pairs. This "negative differentiation" utilizes the negative tail of the distribution heavily. Even following the GELU nonlinearity, an appreciable difference remains in their values as the very negative values are driven close to zero but the less negative values remain. This sign-specific separation concentrates on contexts of functional words and is consistent with the early-layer syntactic scaffolding implicated by our ablations.

Taken together, this single-unit case study shows that Wasserstein neurons can enforce large separations among nearby inputs by pushing both items into the negative region to different degrees—a sign-specific mechanism that targets functional contexts. Having established this intuition at the neuron level, we now scale up: we quantify how common this negative differentiation is across neurons and layers, and how it evolves during training and across model families.

## 6 NEGATIVE DIFFERENTIATION CONCENTRATES EARLY

Having seen negative differentiation in a single unit, we now ask how the sign pattern of differentiated pairs—negative-negative (NN) for two negative pre-activation values, positive-negative (PN) for one of each, and positive-positive (PP) for two positive values—varies across layers and over training. In each layer of Pythia 1.4B, the top $5\%$ entangled neurons are selected by MD, and their top 100 most differentiated pairs out of 1000 are analyzed as before. We label each pair according to the signs of the two output pre-activations.

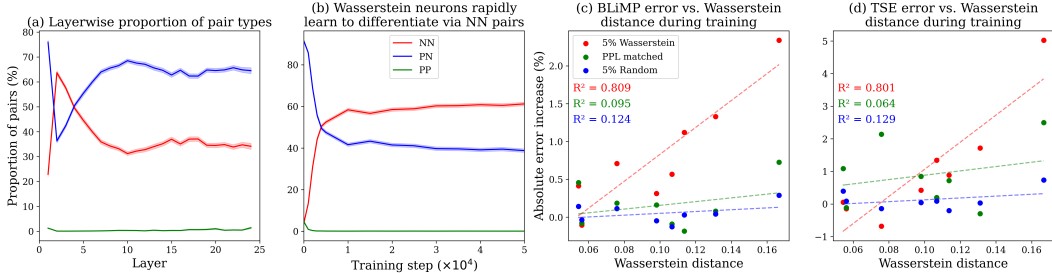

Figure 7: Negative differentiation emerges early and persists across depth in Pythia 1.4B. (a, b) share the same legend and y-axis label. (a) Layerwise composition of the most separated input pairs for the top $5\%$ MD neurons, tracking the top 100 tokens out of 1000 per neuron. Proportion of NN, PN, PP pairs across layers reveals widespread usage of negative differentiation across layers, with the most practitioners belonging to layer 2. (b) In layer 2, the prevalence of NN output pairs in the most differentiated pairs rapidly arises with training in entangled neurons, indicating early specialization into negative differentiation. (c, d) share the same legend and y-axis label. (c) At each checkpoint, the same $5\%$ of Wasserstein neurons are perturbed, and the resulting BLiMP error correlates very strongly to the WD of the cohort at that checkpoint. (d) Analysis from (c) repeated for TSE. For (c, d), neither the random nor the perplexity-matched controls (plotted at the same WD for reference) correlate with error. Data acquired from the top $5\%$ of Wasserstein neurons in Pythia 1.4B. Shaded bands are one standard error of the mean.

We find that NN pairs are highly prevalent in early layers, especially layer 2, but that all layers have a significant proportion of these pairs (at least $30\%$ with the exception of layer 1) while PP pairs are comparatively rare (Figure 7a). This indicates that Wasserstein neurons in Pythia frequently separate similar inputs to negative values of varying degree, rather than always two values of different signs. To better investigate this in layer 2, we tracked the same most entangled neurons over training checkpoints, and measured the properties of the pairs they most differentiated as training progressed. We strikingly find that there is rapid specialization: while NN and PP pairs both occupy a very small fraction of the most differentiated pair type at the onset of training, NN pairs rapidly become prevalent, unlike PP pairs. PN pairs correspondingly decrease in frequency as NN pairs rise (Figure 7b). Thus, negative differentiation is an early emerging and sustained strategy.

To relate this behaviorally to grammar causally rather than by correlation (Figure 2d), at each checkpoint we clamp only the negative pre-activations of $5\%$ of Wasserstein neurons in Pythia

1.4B and measure the resulting accuracy drop on BLiMP and TSE, compared to the random and perplexity-matched controls. Plotting the error increase against the mean WD of the cohort at that checkpoint shows a strong positive relationship: as Wasserstein neurons' distributions become more non-Gaussian, ablation yields larger grammatical damage (Figure 7c, d). Random and perplexity-matched controls, plotted at the same WD for reference, show little to no correlation. Together, these results indicate that as Wasserstein neurons mature, the model increasingly relies upon them for grammar, and so error increases as these neurons' pre-activations diverge from a Gaussian. Additional experiments observing this phenomenon in Llama 3.1 8B can be found in Section A.6. Taken together, our results show that negative pre-activations are mechanistically salient rather than inert: they are leveraged by a sparse set of entangled neurons to differentiate syntax.

## 7 RELATED WORK

### 7.1 ACTIVATION FUNCTIONS

Modern transformers largely replace ReLU with smooth activations such as GELU and SiLU to ease optimization, avoid dead neurons, and empirically improve performance (Hendrycks & Gimpel, 2016; Elfwing et al., 2018; Shazeer, 2020; Lu et al., 2019), with contemporaneous work examining sign-conditional effects in gated MLPs (Gerstner & Schuetze, 2025). Our contribution is to show that models actively leverage the negative pre-activation region beyond simply for training: selectively clamping only negative pre-activations of high-WD neurons impairs perplexity and grammar far beyond random or perplexity-matched controls, revealing a sign-specific computation that prior optimization-centric discussions did not examine.

### 7.2 GRAMMAR ACQUISITION

A long line of work links internal representations to grammatical phenomena using probes, causal tracing, and targeted evaluations (Marvin & Linzen, 2018; Warstadt et al., 2020; Saphra & Lopez, 2018; Tenney et al., 2019; Mueller et al., 2022; Müller-Eberstein et al., 2023; Duan et al., 2025; Chen et al., 2023). Probing studies have shown that syntactic structure is often encoded in middle transformer layers, especially in attention mechanisms (Hewitt & Manning, 2019). Recent approaches with sparse autoencoders recover interpretable, grammar-relevant features from hidden states (Cunningham et al., 2023; Jing et al., 2025; Brinkmann et al., 2025). Together, these studies suggest that grammatical competence emerges early in training and accumulates across depth, and have largely framed syntax as residing in attention patterns or dense hidden subspaces. Our results reveal a complementary avenue through which grammar is computed, especially compared to ReLU models that must necessarily learn grammar through a different mechanism (Sinha et al., 2023). We identify a sign-specific mechanism in the MLP, where early layers use negative differentiation in a small group of entangled neurons, and show that ablating negative pre-activations in these neurons causally degrade grammatical performance, linking emergence, layerwise structure, and behavior at the level of individual neurons.

### 7.3 INTERPRETABILITY

Interpretability in modern machine learning remains difficult in part because many neurons are polysemantic, responding to different features (Arora et al., 2018; Mu & Andreas, 2020; Olah et al., 2020; Goh et al., 2021; Jermyn et al., 2022; Gurnee et al., 2023; Templeton, 2024; Gurnee et al., 2024). A central driver of this is superposition: features are compressed onto shared directions, allowing networks to represent more features than neurons, creating entanglement (Elhage et al., 2022; Lecomte et al., 2023; Adler et al., 2025). Prior work identified an orthogonal form of entanglement in which certain neurons separate highly similar inputs (Sawmya et al., 2025), but only showed that these neurons are sensitive to weight sparsification, without interpreting their role or linking them to syntax. Here, we demonstrate that these Wasserstein neurons specifically leverage the negative pre-activation region to implement syntactic differentiation, an aspect underexplored in analyses that implicitly treat positive pre-activations as the sole carrier of signal.

## 8 CONCLUSION AND DISCUSSION

We have shown that the negative pre-activation region of smooth nonlinearities is actively used to support syntax in large language models. This usage is most readily observable in Wasserstein neurons. Such neurons emerge and stabilize early in training, tracking the development of grammatical competence. Causal, sign-specific ablations that zero only their negative pre-activations sharply disrupt BLiMP and TSE performance with comparatively less impact on non-grammatical tasks, whereas perplexity-matched ablations of many more low-WD neurons largely spare grammar but cause greater degradation on other benchmarks, yielding a double dissociation that demonstrates the complexity of the negative pre-activation space. Together, these results indicate that negative pre-activations are not an inert byproduct of GELU or SiLU, but a functional substrate that certain neurons leverage to implement grammar. This reframes common ReLU-era intuitions that equate "activity" with positive pre-activations and underscores that interpretability methods must attend to the full activation landscape, including negative regions where crucial computation can reside.

### ACKNOWLEDGMENTS

The authors would like to extend their gratitude to many members of the interpretability community for fruitful discussions and insightful comments. In particular, we thank Ben Wu, Georg Lange, Can Rager, Josh Engels, and Alessandro Stolfo for their thoughtful input. We are also grateful to Jacob Andreas, Dan Alistarh, Dan Gutfreund, Alexandre Marques, Tony Wang, and Thomas Athey for providing valuable feedback and constructive guidance as the project developed.

This project was supported by an MIT-IBM Watson AI Lab grant and an NIH Brains CONNECTS U01 grant. Computational resources were provided by Red Hat AI and AMD's AI & HPC Fund.

Finally, we dedicate this work to the memory of Shashata Sawmya, whose friendship, insight, and curiosity are deeply missed.

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

# A APPENDIX

## A.1 VALIDATING THE WASSERSTEIN DISTANCE AS A PROXY OF MAPPING DIFFICULTY

### A.1.1 WASSERSTEIN DISTANCE VS. MAPPING DIFFICULTY FOR EVERY NEURON

To demonstrate the association of a neuron's Wasserstein distance and its mapping difficulty, we calculate every neuron's WD and MD in each gate projection layer in Llama 3.1 8B. In each layer, the two metrics track one another closely.

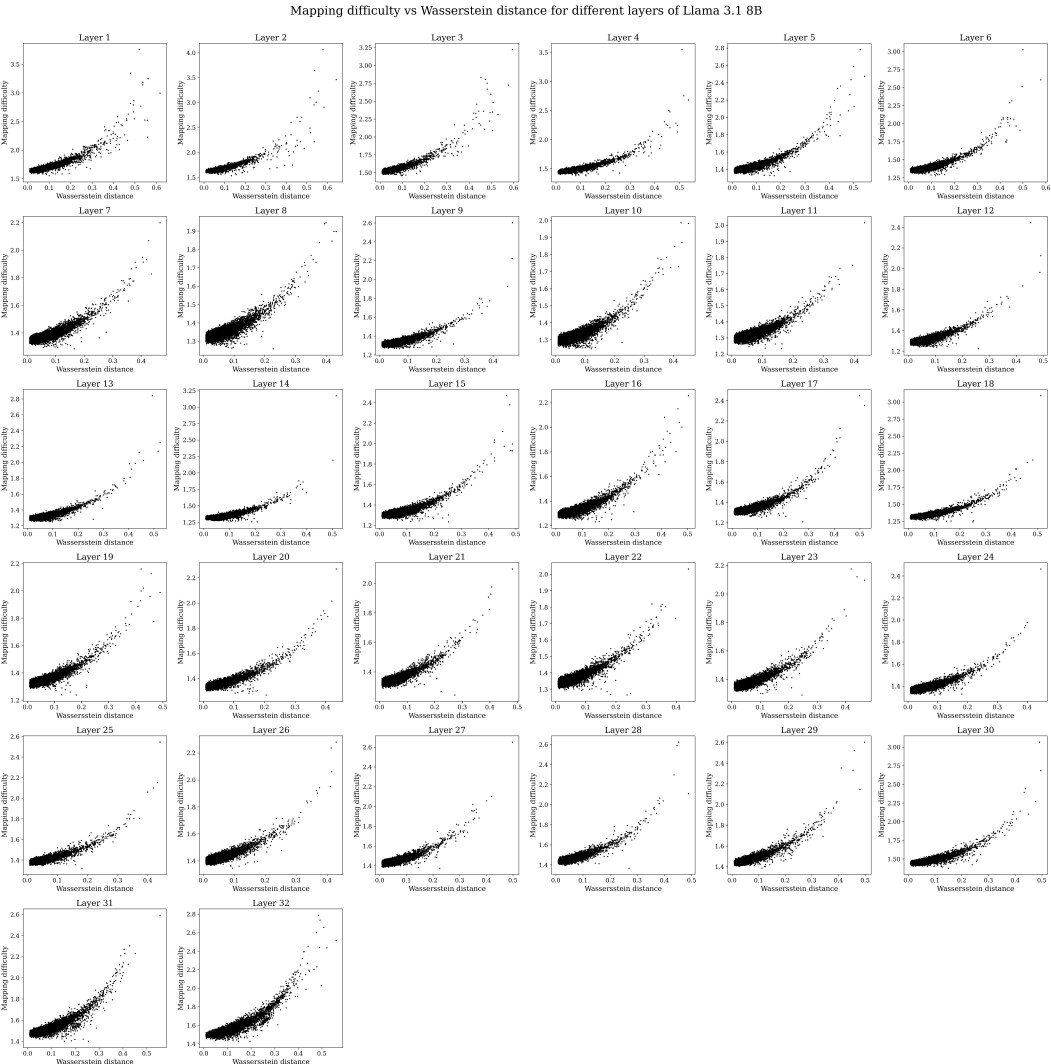

Figure A1: The Wasserstein distance of a neuron closely tracks its mapping difficulty. Neurons from every gate projection in Llama 3.1 8B. Data used to calculate metrics from WikiText 2.

### A.1.2 WASSERSTEIN DISTANCE MORE CLOSELY ASSOCIATES WITH MAPPING DIFFICULTY THAN WITH ASYMMETRY AND KURTOSIS

To investigate the possible confounds that skewness or kurtosis may cause in the calculation of WD, we calculate the WD, asymmetry $|\gamma_1|$, and excess kurtosis $|\kappa|$ of the output distribution for each neuron in each gate projection in Llama 3.1 8B. We then compare these metrics between themselves and with MD. Because our analysis utilizes the most entangled neurons, we calculate the Jaccard index between the top $1\%$ of neurons ranked by each metric and measure agreement, with a Jaccard index of 0 indicating no agreement and 1 indicating complete agreement. Across all layers, the agreement of the top $1\%$ of neurons as calculated by WD and MD is the highest, compared to WD and asymmetry, WD and kurtosis, MD and asymmetry, and MD and kurtosis (Table A1).

Table A1: Jaccard index of the top $1\%$ of neurons as ranked by Wasserstein distance, mapping difficulty, asymmetry, and kurtosis in each gate projection layer of Llama 3.1 8B. Greatest values in each layer are bolded.

| Layer | $J(WD, MD)$ | $J(WD, |\gamma_1|)$ | $J(WD, |\kappa|)$ | $J(MD, |\gamma_1|)$ | $J(MD, |\kappa|)$ |
|---|---|---|---|---|---|
| 1 | **0.713** | 0.294 | 0.083 | 0.222 | 0.051 |
| 2 | **0.713** | 0.452 | 0.217 | 0.349 | 0.153 |
| 3 | **0.810** | 0.474 | 0.202 | 0.437 | 0.182 |
| 4 | **0.799** | 0.497 | 0.283 | 0.452 | 0.254 |
| 5 | **0.810** | 0.474 | 0.197 | 0.459 | 0.192 |
| 6 | **0.833** | 0.529 | 0.238 | 0.513 | 0.233 |
| 7 | **0.810** | 0.423 | 0.207 | 0.423 | 0.202 |
| 8 | **0.673** | 0.474 | 0.222 | 0.474 | 0.254 |
| 9 | **0.733** | 0.521 | 0.192 | 0.563 | 0.212 |
| 10 | **0.702** | 0.490 | 0.202 | 0.529 | 0.243 |
| 11 | **0.755** | 0.505 | 0.227 | 0.505 | 0.238 |
| 12 | **0.692** | 0.474 | 0.254 | 0.482 | 0.277 |
| 13 | **0.776** | 0.482 | 0.233 | 0.497 | 0.254 |
| 14 | **0.673** | 0.521 | 0.277 | 0.513 | 0.294 |
| 15 | **0.744** | 0.437 | 0.207 | 0.490 | 0.222 |
| 16 | **0.799** | 0.409 | 0.227 | 0.368 | 0.197 |
| 17 | **0.787** | 0.416 | 0.172 | 0.416 | 0.187 |
| 18 | **0.787** | 0.416 | 0.192 | 0.409 | 0.172 |
| 19 | **0.733** | 0.388 | 0.172 | 0.368 | 0.177 |
| 20 | **0.744** | 0.444 | 0.177 | 0.416 | 0.167 |
| 21 | **0.682** | 0.416 | 0.144 | 0.375 | 0.153 |
| 22 | **0.755** | 0.474 | 0.197 | 0.423 | 0.187 |
| 23 | **0.702** | 0.444 | 0.182 | 0.452 | 0.207 |
| 24 | **0.776** | 0.474 | 0.197 | 0.474 | 0.212 |
| 25 | **0.733** | 0.505 | 0.207 | 0.452 | 0.217 |
| 26 | **0.755** | 0.474 | 0.233 | 0.459 | 0.217 |
| 27 | **0.755** | 0.505 | 0.217 | 0.529 | 0.249 |
| 28 | **0.810** | 0.529 | 0.254 | 0.529 | 0.254 |
| 29 | **0.755** | 0.505 | 0.217 | 0.490 | 0.222 |
| 30 | **0.755** | 0.388 | 0.187 | 0.402 | 0.192 |
| 31 | **0.776** | 0.324 | 0.109 | 0.288 | 0.092 |
| 32 | **0.810** | 0.243 | 0.092 | 0.197 | 0.067 |

We also show the visual correlation between each the WD and MD, WD and asymmetry, WD and kurtosis, MD and asymmetry, and MD and kurtosis for representative gate projection layers in Llama 3.1 8B. We find that MD and WD track each other more closely than either metric with either asymmetry or kurtosis.

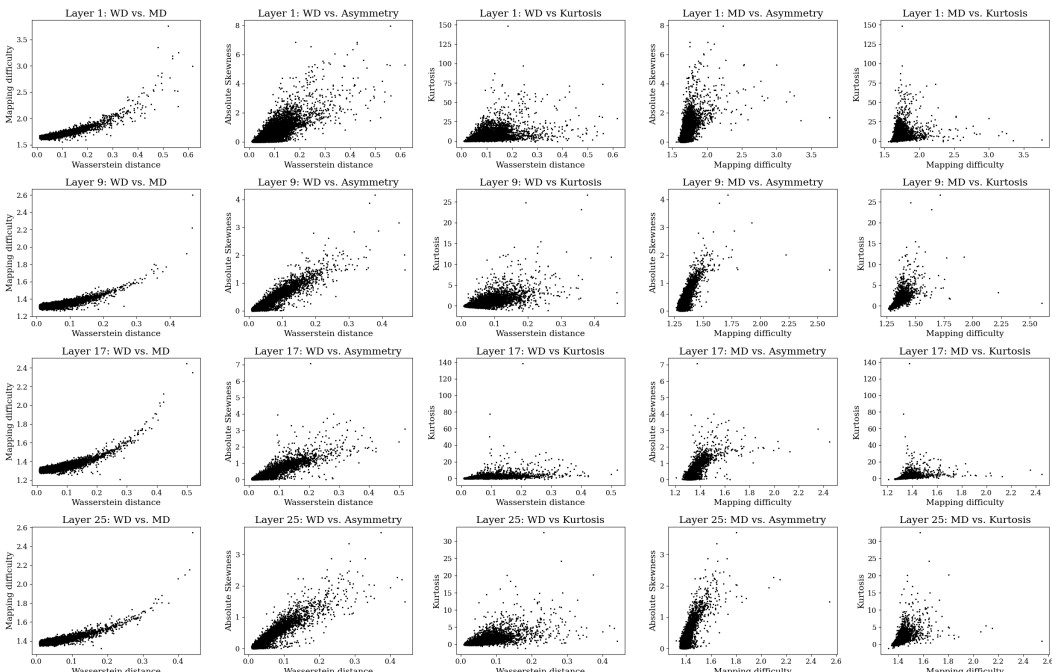

Figure A2: Wasserstein distance and mapping difficulty correlate more closely than asymmetry or kurtosis. Neurons from specified gate projection layers in Llama 3.1 8B. Data used to calculate metrics from WikiText 2.

### A.1.3 USING MAPPING DIFFICULTY DIRECTLY RATHER THAN THE WASSERSTEIN DISTANCE PROXY YIELDS THE SAME QUALITATIVE RESULT IN CAUSAL ABLATIONS

To validate the choice of WD as a proxy of MD, we also directly use MD as a selection criteria for the most entangled neurons, rather than WD. Repeating the experiments in Figure 3 for Llama 3.1 8B yields the same qualitative result: the negative pre-activation is highly sensitive for these neurons and is implicated directly in grammatical function.

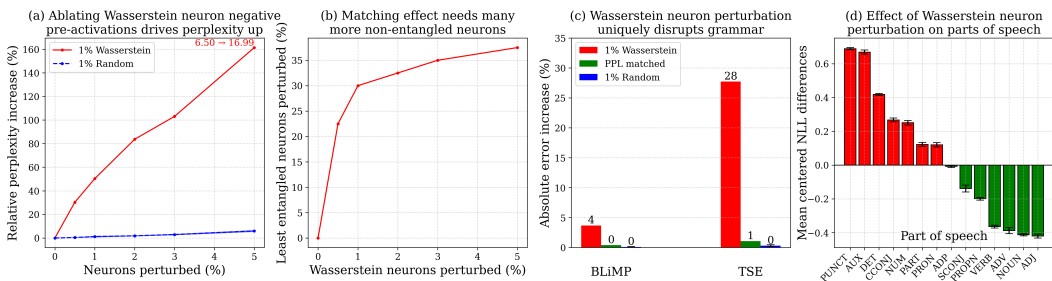

Figure A3: Sign-specific perturbation of Wasserstein neurons disproportionately harms grammar when using MD instead of WD as the selection metric as well. (a) Perplexity increases when clamping only the negative pre-activations of the top-MD neurons; random controls are much smaller. Number indicates starting and ending perplexity in absolute terms. (b) Matching the perplexity increase from perturbing the entangled fraction of neurons requires an order of magnitude more non-entangled units. (c) Perturbing Wasserstein neurons, as chosen by MD, uniquely impacts grammatical capabilities, even when compared to the perplexity-matched control. The top $1\%$ Wasserstein neurons in each layer, as chosen by MD, were perturbed for the benchmark. The least entangled $30\%$ of neurons, as chosen by MD, were used as the perplexity-matched control. (d) At a per token resolution of the WikiText 2 validation dataset, tokens associated with syntactical scaffolding experience a much higher surprisal for the $1\%$ Wasserstein perturbation, as chosen by MD, compared to the perplexity matched control. NLL differences were mean shifted by the global difference. Randomly sampled controls were acquired over ten trials. Error bars indicate one standard error of the mean. Data from Llama 3.1 8B.

## A.2 RAW BENCHMARK SCORES

Table A2: Model performance on BLiMP and TSE with the applied perturbations. Data used in Figure 3c. Randomly sampled controls were acquired over ten trials. Lowest value in bold.

| Model | Benchmark | Baseline | 1% Wasserstein | PPL Matched | 1% Random |
|---|---|---|---|---|---|
| Llama 3.1 8B | BLiMP | $81.2 \pm 0.2$ | $\mathbf{77.5 \pm 0.2}$ | $81.5 \pm 0.2$ | $81.1 \pm 0.2$ |
| | TSE | $84.4 \pm 0.1$ | $\mathbf{55.2 \pm 0.1}$ | $85.9 \pm 0.1$ | $84.1 \pm 0.1$ |
| Mistral 7B v0.3 | BLiMP | $83.3 \pm 0.1$ | $\mathbf{77.9 \pm 0.2}$ | $82.8 \pm 0.1$ | $83.2 \pm 0.1$ |
| | TSE | $84.6 \pm 0.1$ | $\mathbf{54.6 \pm 0.1}$ | $87.2 \pm 0.1$ | $84.5 \pm 0.1$ |
| Qwen3 8B Base | BLiMP | $82.5 \pm 0.1$ | $\mathbf{80.6 \pm 0.2}$ | $81.3 \pm 0.2$ | $82.4 \pm 0.1$ |
| | TSE | $84.8 \pm 0.1$ | $\mathbf{78.6 \pm 0.1}$ | $82.4 \pm 0.1$ | $84.7 \pm 0.1$ |

Table A3: Llama 3.1 8B performance on a suite of non-grammar benchmarks with the applied perturbations. Data used in Figure 4. Randomly sampled controls were acquired over ten trials. Lowest value in bold.

| Benchmark | Baseline | 1% Wasserstein | PPL Matched | 1% Random |
|---|---|---|---|---|
| ARC Challenge | $53.4 \pm 1.5$ | $48.6 \pm 1.5$ | $\mathbf{38.1 \pm 1.4}$ | $53.3 \pm 1.5$ |
| ARC Easy | $81.1 \pm 0.8$ | $76.2 \pm 0.9$ | $\mathbf{60.4 \pm 1.0}$ | $80.7 \pm 0.8$ |
| BoolQ | $82.1 \pm 0.7$ | $74.9 \pm 0.8$ | $\mathbf{67.0 \pm 0.8}$ | $81.9 \pm 0.7$ |
| HellaSwag | $78.9 \pm 0.4$ | $75.3 \pm 0.4$ | $\mathbf{69.4 \pm 0.5}$ | $78.6 \pm 0.4$ |
| PIQA | $81.2 \pm 0.9$ | $78.8 \pm 1.0$ | $\mathbf{76.6 \pm 1.0}$ | $81.4 \pm 0.9$ |
| SciQ | $94.6 \pm 0.7$ | $95.4 \pm 0.7$ | $\mathbf{85.3 \pm 1.1}$ | $94.5 \pm 0.7$ |
| TruthfulQA | $45.2 \pm 1.4$ | $\mathbf{39.8 \pm 1.4}$ | $42.9 \pm 1.5$ | $46.0 \pm 1.4$ |
| WinoGrande | $73.9 \pm 1.2$ | $70.4 \pm 1.3$ | $\mathbf{66.3 \pm 1.3}$ | $74.7 \pm 1.2$ |

Table A4: Mistral 7B v0.3 performance on a suite of non-grammar benchmarks with the applied perturbations. Data used in Figure A4. Randomly sampled controls were acquired over ten trials. Lowest value in bold.

| Benchmark | Baseline | 1% Wasserstein | PPL Matched | 1% Random |
|---|---|---|---|---|
| ARC Challenge | $52.3 \pm 1.5$ | $52.4 \pm 1.5$ | $\mathbf{43.5 \pm 1.4}$ | $52.7 \pm 1.5$ |
| ARC Easy | $78.2 \pm 0.8$ | $77.0 \pm 0.9$ | $\mathbf{65.2 \pm 1.0}$ | $78.5 \pm 0.8$ |
| BoolQ | $82.1 \pm 0.7$ | $79.8 \pm 0.7$ | $\mathbf{71.4 \pm 0.8}$ | $81.9 \pm 0.7$ |
| HellaSwag | $80.4 \pm 0.4$ | $79.4 \pm 0.4$ | $\mathbf{72.5 \pm 0.4}$ | $80.3 \pm 0.4$ |
| PIQA | $82.3 \pm 0.9$ | $81.9 \pm 0.9$ | $\mathbf{76.8 \pm 1.0}$ | $81.9 \pm 0.9$ |
| SciQ | $94.0 \pm 0.8$ | $95.1 \pm 0.7$ | $\mathbf{83.7 \pm 1.2}$ | $94.2 \pm 0.7$ |
| TruthfulQA | $42.6 \pm 1.4$ | $\mathbf{41.4 \pm 1.4}$ | $46.1 \pm 1.5$ | $42.7 \pm 1.4$ |
| WinoGrande | $73.9 \pm 1.2$ | $71.1 \pm 1.3$ | $\mathbf{61.6 \pm 1.4}$ | $73.6 \pm 1.2$ |

Table A5: Qwen3 8B Base performance on a suite of non-grammar benchmarks with the applied perturbations. Data used in Figure A5. Randomly sampled controls were acquired over ten trials. Lowest value in bold.

| Benchmark | Baseline | 1% Wasserstein | PPL Matched | 1% Random |
|---|---|---|---|---|
| ARC Challenge | $56.9 \pm 1.4$ | $\mathbf{51.5 \pm 1.5}$ | $54.2 \pm 1.5$ | $56.8 \pm 1.4$ |
| ARC Easy | $80.0 \pm 0.8$ | $\mathbf{75.5 \pm 0.9}$ | $78.0 \pm 0.8$ | $80.4 \pm 0.8$ |
| BoolQ | $83.0 \pm 0.7$ | $\mathbf{81.9 \pm 0.7}$ | $82.2 \pm 0.7$ | $83.2 \pm 0.7$ |
| HellaSwag | $78.6 \pm 0.4$ | $78.8 \pm 0.4$ | $\mathbf{74.7 \pm 0.4}$ | $78.6 \pm 0.4$ |
| PIQA | $79.3 \pm 0.9$ | $79.1 \pm 0.9$ | $\mathbf{78.6 \pm 1.0}$ | $79.3 \pm 0.9$ |
| SciQ | $96.1 \pm 0.6$ | $96.0 \pm 0.6$ | $\mathbf{94.7 \pm 0.7}$ | $95.9 \pm 0.6$ |
| TruthfulQA | $52.3 \pm 1.5$ | $\mathbf{50.9 \pm 1.5}$ | $54.8 \pm 1.5$ | $52.6 \pm 1.5$ |
| WinoGrande | $72.8 \pm 1.2$ | $69.5 \pm 1.3$ | $\mathbf{68.6 \pm 1.3}$ | $71.9 \pm 1.3$ |

## A.3 Non-grammatical benchmarks for additional models

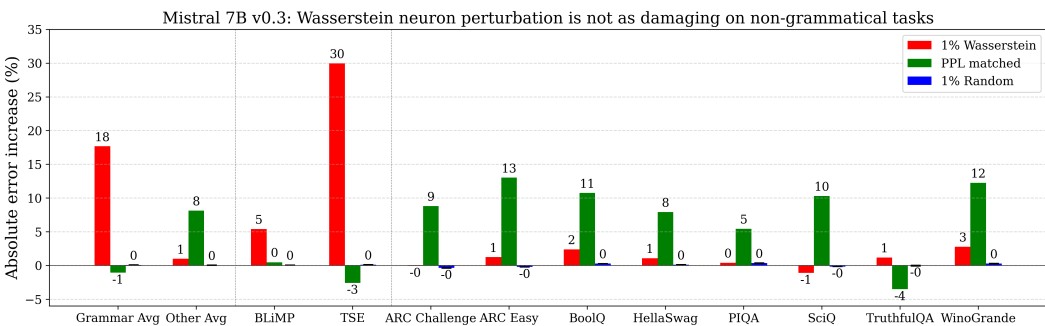

Figure A4: Non-grammatical abilities are less harmed by Wasserstein neuron perturbation in Mistral 7B v0.3. All benchmarks were run in 0-shot. Randomly sampled controls were acquired over ten trials. Error bars indicate one standard error of the mean. Raw scores and CI's are in Tables A2, A4.

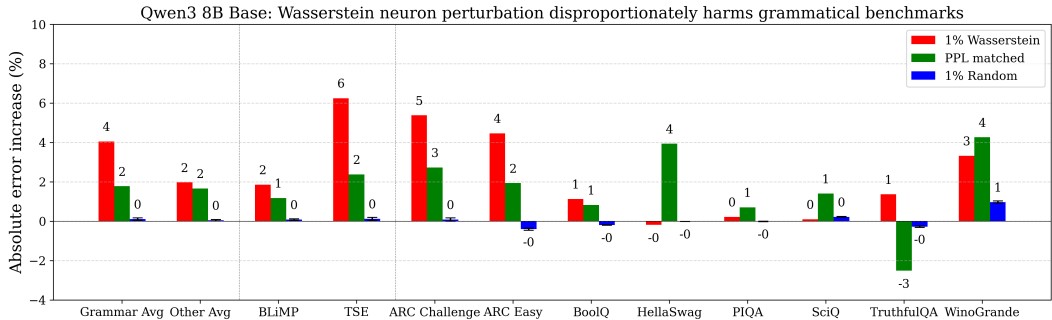

Figure A5: Non-grammatical abilities are comparatively less harmed by Wasserstein neuron perturbation in Qwen3 8B Base. Although the Wasserstein neuron perturbation causes more damage in both grammatical and non-grammatical tasks, it still causes more relative and absolute damage in the former. All benchmarks were run in 0-shot. Randomly sampled controls were acquired over ten trials. Error bars indicate one standard error of the mean. Raw scores and CI's are in Tables A2, A5.

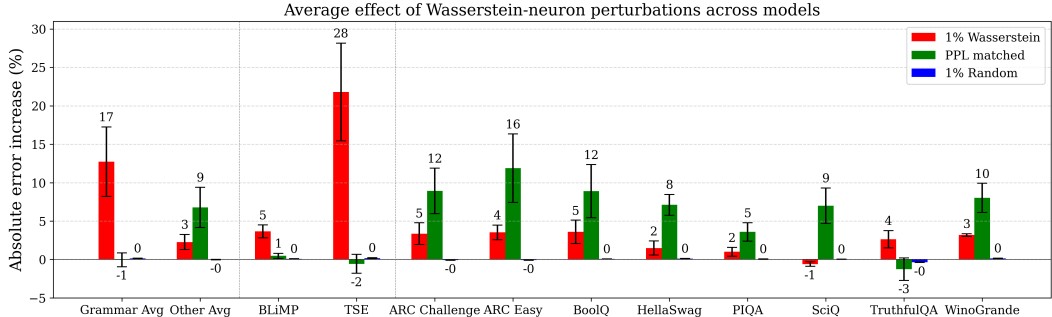

Figure A6: Average effect across Llama 3.1 8B, Mistral 7B v0.3, and Qwen3 8B Base. While individual models show varying degrees of grammar selectivity, on average, grammar benchmarks are substantially more sensitive to targeted Wasserstein neuron perturbations, whereas non-grammatical benchmarks are more sensitive to broad low-WD ablations at matched perplexity. All benchmarks were run in 0-shot. Randomly sampled controls were acquired over ten trials per model. Error bars indicate one standard error of the mean. Raw scores and CI's are in Tables A2, A3, A4, A5.

### A.4 ADDITIONAL ACTIVATION FUNCTION ABLATIONS REVEAL IMPORTANCE OF THE NEGATIVE SIGN ITSELF

We further study additional modifications to the effective activation function and therefore the pre-activation space, revealing the importance of negative sign itself rather than magnitude. We compare four ablations:

1. zeroing negative pre-activations, as was done in Section 3
   $y = \text{SiLU}(x)$ if $x > 0, y = 0$ otherwise

2. setting positive pre-activations to use the ReLU activation function rather than SiLU
   $y = \text{SiLU}(x)$ if $x < 0, y = x$ otherwise

3. zeroing positive pre-activations
   $y = \text{SiLU}(x)$ if $x < 0, y = 0$ otherwise

4. flipping the sign of negative post-activations
   $y = \text{SiLU}(x)$ if $x > 0, y = -\text{SiLU}(x)$ otherwise

For each control, as in Section 3, we modify the activation function for the top-WD neurons in each layer and compare that to modifying the activation function in the same way for an equal number of randomly chosen neurons in each layer.

As an initial confirmation, the curvature of the SiLU in the positive is not particularly salient, as replacing that with ReLU yielded very little change to model performance. By contrast, zeroing the positive signal of Wasserstein neurons is substantially more damaging than applying the same intervention to random neurons and also more damaging than zeroing the negative signal, consistent with the positive branch carrying the bulk of the model's overall information.

Strikingly, reversing the sign of negative post-activations yields substantial model damage, moreso than even zeroing out positive activations at 2% ablation. Indeed, with just a reversal of the negative signs of the top 2% WD neurons, perplexity climbs from 6.50 to 2277. Crucially, this intervention preserves the magnitude of the negative responses and thereby largely preserves their contribution to RMS/LayerNorm statistics, while inverting only the sign. Together, these controls show that the sign of negative activations in Wasserstein neurons carries essential information, and inverting it is more harmful than removing it altogether.

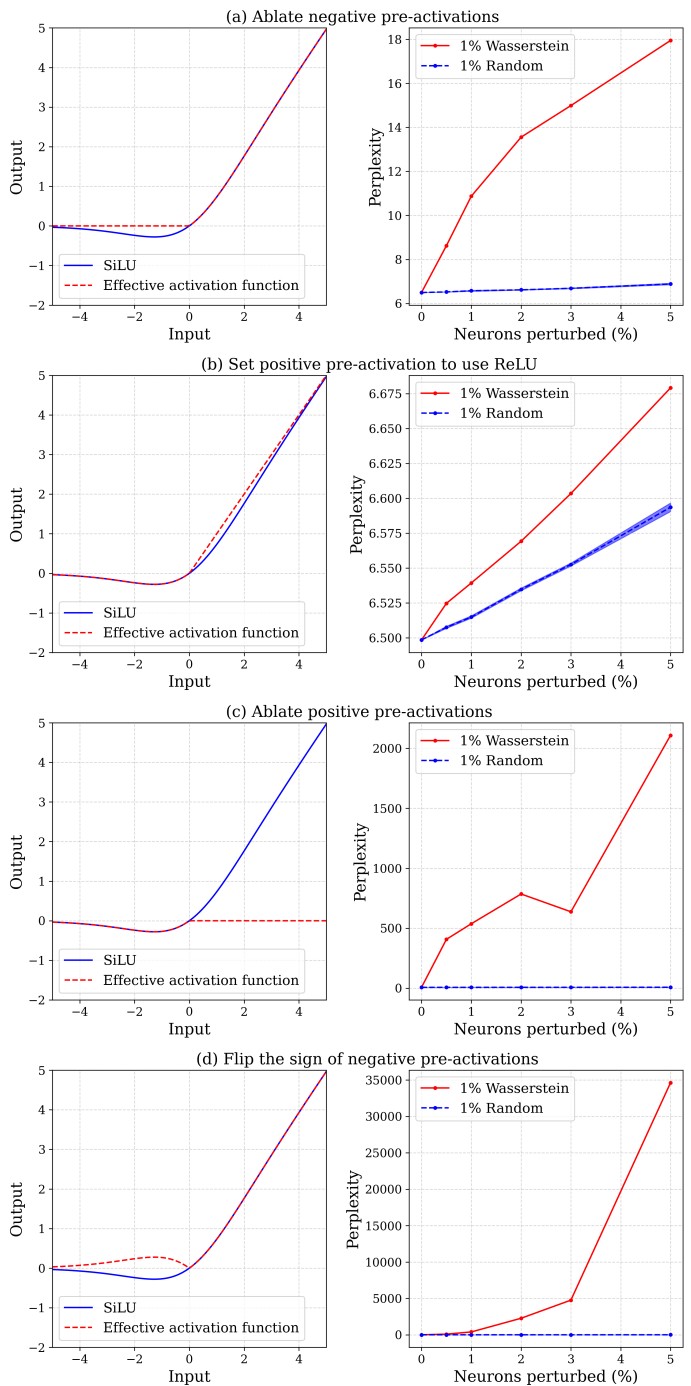

Figure A7: Additional controls reveal importance of negative sign itself rather than just its magnitude. For each control, as in Section 3, we modify the activation function for the top-WD neurons in each layer and compare that to modifying the activation function in the same way for an equal number of randomly chosen neurons in each layer. For each modification, the effective activation function is shown in the left column, and the effect on model perplexity on the WikiText 2 validation dataset is shown in right column. (a) Zeroing negative pre-activations, as was done in Section 3. (b) Setting positive pre-activations to use the ReLU activation function rather than SiLU. (c) Zeroing positive pre-activations. (d) Flipping the sign of negative post-activations.

### A.5 FURTHER CHARACTERIZATION OF WASSERSTEIN NEURONS AND PROSPECTS FOR AUTOMATION

To probe how interpretable the differentiation of tokens is among Wasserstein neurons, we extend the analysis of Figure 6 to additional neurons in the second up projection layer in Pythia 1.4B. For each neuron, we collect the top output pairs that have been mapped the furthest given their input distance, and observe the underlying tokens that these inputs correspond to. Doing so reveals further clear, neuron-specific patterns. For example, neuron 1168 strongly separates the token "and" from a variety of other tokens, neuron 4093 preferentially differentiates past-tense verb forms, neuron 4457 separates adpositions from determiners, neuron 4606 distinguishes punctuation from other tokens, neuron 5776 isolates numerical tokens from other vocabulary items, and neuron 6984 is selective for determiners (Figure A8).

These qualitative patterns are reflected quantitatively in the POS statistics of the top differentiated pairs. Considering the top 50 token pairs per neuron, we find that specific POS categories (determiners, adpositions, punctuation, numerals, verbs) are heavily enriched among the differentiated tokens compared to their base rates in WikiText 2. This suggests that many Wasserstein neurons implement relatively coherent, fine-grained syntactic distinctions. While we stop short of building a fully automated mapping from neurons to syntactic features, these POS enrichment statistics provide a natural starting point for such a system in future work.

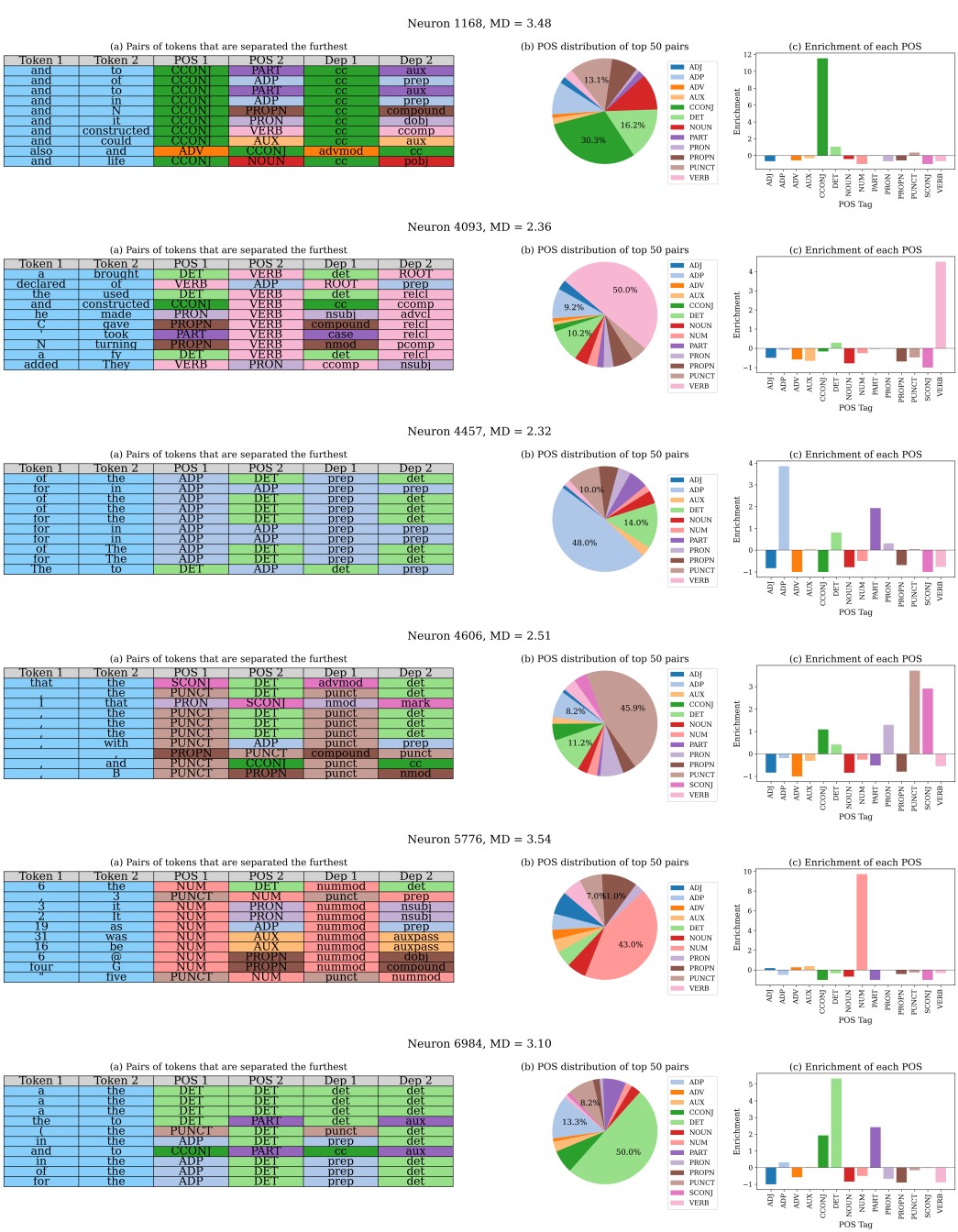

Figure A8: Additional examples of interpretable Wasserstein neurons based on the tokens they differentiate. For each neuron, the top 10 pairs of tokens that are most differentiated are visualized in (a). The distribution of each POS within the top 50 pairs that are most differentiated is shown in (b). These distributions are compared to the underlying POS distribution for the entire WikiText 2 dataset, and the relative enrichment is visualized in (c). The relative enrichment for each POS is calculated as the ratio between the proportion of that POS in the top 50 pairs and the proportion of that POS in the WikiText 2 dataset as a whole, minus 1.

### A.6 Layerwise negative differentiation predicts grammatical error in Llama

We repeat the pair analysis and sign-specific ablations presented in Section 6 for Llama 3.1 8B. In this case, we use the top $0.5\%$ MD of neurons, with 50 out of 1000 pairs each to increase specificity. Compared to Pythia, Llama exhibits a lower overall rate of NN pairs, with PN dominating across depth. Nevertheless, early layers still show an elevated NN proportion compared to later layers (Figure A9a). To test whether this residual NN usage is behaviorally meaningful, we perturbed the top $1\%$ Wasserstein neurons one layer at a time and measured the induced error on BLiMP and TSE.

Across layers, the fraction of NN pairs in the most separated set positively correlates with error under the perturbation, while PN and PP proportions show weak or opposite trends. Both BLiMP and TSE show this pattern (Figure A9b, c). Thus, even in an architecture that uses NN differentiation less overall, layers that rely on it suffer the most when negative pre-activations are suppressed.

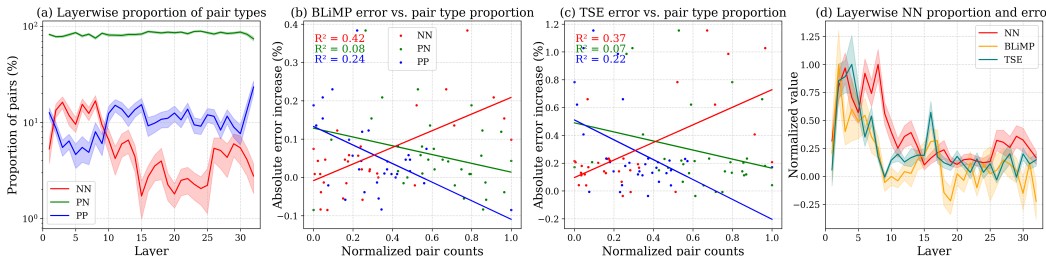

Figure A9: Llama uses less negative differentiation overall, yet early-layer NN still predicts grammatical fragility. (a) Layerwise composition of the most-separated pairs (top 50 of 1000 per neuron) for the top $0.5\%$ MD neurons in Llama. PN dominates, but NN remains elevated in early layers. Log scale used for y-axis to better show differences in trends. (b, c) share the same legend. (b) The induced error in BLiMP from a layerwise $1\%$ Wasserstein neuron negative pre-activation ablation correlates with the proportion of NN pairs in that layer. (c) Analysis from (b) repeated for TSE. (d) Normalized overlay of NN proportion, BLiMP error, and TSE error over layer depth highlights a shared early layer peak, indicating that layers with more NN differentiation are those whose performance degrades most under negative clamping. Shaded regions are one standard error of the mean.

Finally, overlaying the normalized NN proportion with the normalized BLiMP and TSE error reveals closely aligned peaks in the earliest layers (Fig. A9d). This alignment supports a general picture: negative differentiation is an early-layer mechanism that downstream computation depends on, even when its global prevalence is modest.

However, the interpretation of NN differentiation is limited by a structural confound in our ablation method. Because our intervention zeroes only negative pre-activations, it disproportionately collapses NN pairs relative to PN or PP pairs. As a result, correlations between the prevalence of NN pairs and grammatical vulnerability under ablation may partly reflect the mechanics of the intervention itself, rather than intrinsic grammatical importance. We therefore treat this analysis as suggestive rather than causal, and focus our main claims on the sign-specific causal ablations. Our sign-flip experiment (Section A.4) partly mitigates this concern by modifying the sign of negative activations while preserving magnitude, producing even larger degradation. Nonetheless, fully controlled comparisons between NN, PN, and PP specific perturbations remain an open direction.

