# OpenReview forum: "Negative Pre-activations Differentiate Syntax"
_ICLR.cc/2026/Conference — ICLR 2026 Poster_

### Official Review · Reviewer_LJm9 · 2025-10-21

**Soundness:** 3
**Presentation:** 2
**Contribution:** 2
**Rating:** 2
**Confidence:** 3

**Summary:**

This paper examines the role of so-called 'Wasserstein neurons' in transformer networks, emphasising their use of negative activations to encode syntactic information. Building upon prior work on Wasserstein neurons, the authors analyse how clamping these negative activations to zero affects model performance.
Experiments demonstrate that eliminating negative activations in Wasserstein neurons significantly degrades model performance, whereas the same intervention has minimal impact on non-Wasserstein neurons.

**Strengths:**

The concept of Wassestein neurons is interesting, and this work tries to explain why they are important.

Important plots have error bars.

The importance of Wasserstein neurons for grammar in LLMs is rigorously shown.

**Weaknesses:**

1) While the findings are interesting, the work largely extends ideas already presented in “Wasserstein Distances, Neuronal Entanglement, and Sparsity.”
I understand the main experimental results as follows: Suppressing activations in neurons identified as important (from previous work) harms model performance more than suppressing other neurons, which is intuitive and expected from the original discussion.

2) The part of specifically attributing grammar to these neurons is not convincing.
Is it possible that grammar is simply the first thing that is lost when a model degenerates?
How does performance decrease on non-grammar benchmarks when disturbing Wasserstein-Neurons? Perplexity indicates that changes could be large, too, which would contradict a specific attribution.

**Minor remarks:**

Figure 3 states: "(a,b) share the same legend." However, they have different x and y labels and different numbers on the axis.

The text has an abundance of -- [some text] -- mid-sentences, which disturbs the flow of reading.

**Questions:**

Specifically, the "CONCLUSION AND DISCUSSION" part reads to me like all it could be logically derived from the statement:
"Wasserstein neurons are important and modifying them disrupts model performance."
Could the authors elaborate on key findings that are rigorously analyzed with e.g., ablation studies that are not in very close relation to this?

---

> ### Author Response · Authors · 2025-11-21
> **Response to Reviewer LJm9 (1/4)**
>
> Thank you for your thoughtful and detailed feedback. We have strengthened the paper’s presentation accordingly and clarified the novelty of our contributions, since our original framing may have unintentionally obscured the key conceptual point.
>
> The central contribution of our paper is not simply that Wasserstein neurons matter, but rather an investigation into negative pre-activations themselves, examining whether their presence is as inert as is often assumed. Specifically, we show that the negative pre-activation region of a very small set of neurons implements a specific, sign-dependent computation that is casually necessary for grammatical competence.
>
> This is new and orthogonal to the prior work on Wasserstein neurons, “Wasserstein Distances, Neuronal Entanglement, and Sparsity.” That work introduced WD as a scoring metric, characterizing certain neurons through statistical properties of their output distributions and showing that they were sensitive to weight sparsity. However, that work did not investigate what computation these neurons perform, nor did it examine the negative pre-activation region, which is often assumed to be uninformative even in smooth-activation models. In our work, Wasserstein neurons serve only as a tool, a way to tractably select a small subset of neurons whose negative activations exhibit non-Gaussian structure and are thus plausible candidates for analysis. We now make this framing more clear (L66).
>
> To better show our contributions, we have reorganized the introduction and abstract. We now begin by motivating the question we seek to answer: smooth-activation models allow graded negative pre-activations, but are these values used for anything, or are they effectively ignored? Now, only after raising this mechanistic question do we introduce Wasserstein neurons as a practical way to isolate where this negative region structure is concentrated. We have also more explicitly highlighted our unique contributions in the related works section (L468).
>
> We feel that this new framing highlights our contributions much more significantly, and we appreciate the reviewer’s feedback that led to this change. Below, we address the specific weaknesses.

---

> ### Author Response · Authors · 2025-11-21
> **Response to Reviewer LJm9 (2/4)**
>
> ## Response to weaknesses
>
> ### 1. Framing of the contribution
>
> Thank you for pointing out that the framing of our results in the original submission may have inadvertently caused some of our central contributions to be overlooked. Our contribution is stronger than "suppressing activations in neurons identified as important (from previous work) harms model performance more than suppressing other neurons."
>
> Instead, the paper makes several new, mechanistically specific points, which we separate out as either present in the original submission, or newly included in the rebuttal to strengthen the core claims already made.
>
> **Major contributions already present in the original submission**
>
> 1. *Sign specific negative computation is causally necessary for grammar.* Zeroing only the negative pre-activations, a region generally considered functionally inert, in just 1% of entangled neurons causes outsized increases in perplexity and specifically grammar. Matching the perplexity increase by zeroing out many more neurons does not yield similar effects on grammatical damage (Figure 3c).
>
> 2. *Localization of damage to syntactic scaffolding tokens.* Token-level surprisal analysis shows that excess error from Wasserstein negative pre-activation ablations concentrates on determiners, auxiliaries, particles, and punctuation, not on nouns or adjectives (Figure 3d).
>
> 3. *Layerwise causal structure.* Early MLP layers are most critical for grammar, with local perturbations in that region causing largest BLiMP/TSE degradation. Cumulative ablations also produce monotonic error growth across depth (Figure 5).
>
> 4. *Link to training dynamics.* The same entangled cohort emerges early during pretraining, stabilizes quickly, and its mean WD correlates tightly with TSE accuracy across checkpoints (Figure 2). When we ablate only their negative pre-activations at each checkpoint, the resulting grammatical error increases with the cohort’s WD, whereas random and perplexity-matched controls show little to no relationship (Figure 7c, d). This indicates that as these neurons acquire their characteristic non-Gaussian structure, the model increasingly relies on their negative-region computation for grammatical behavior.
>
> 5. *Neuron-level mechanism: negative differentiation.* Individual entangled neurons map similar inputs to different depths of negativity, separating syntactic tokens via negative-negative output pairs (Figure 7, A6).
>
> **Additional contributions added in the rebuttal**
>
> 1. *Grammar vs non-grammar double dissociation.* Using eight non-grammatical benchmarks (ARC Challenge, ARC Easy, BoolQ, HellaSwag, PIQA, SciQ, TruthfulQA, WinoGrande), we find the inverse pattern to grammatical benchmarks: the perplexity-matched low-WD ablations in the negative pre-activations harm performance on these tasks far more than top-WD ablations. Together with the grammar results, this forms a clear double dissociation between syntax-specific and general capacity (Figure 4). We provide more details below as well.
>
> 2. *Sign-specificity beyond magnitude.* New ablations show that flipping the sign of negative post-activations (preserving magnitude and LayerNorm statistics) causes worse degradation than zeroing either negative or positive pre-activations, confirming that the sign itself carries functional information (Section A3).
>
> 3. *Expanded neuron level evidence showing syntactic specialization at scale.* Building on the original single-neuron case study (Figure 6), we extended the most differentiated token pair analysis across many Wasserstein neurons. For each neuron, we extracted the top separated token pairs and evaluated their POS composition. We observe consistent enrichment for distinct syntactic categories. Some neurons predominantly differentiate determiners vs. adpositions, others past-tense verbs, and others others punctuation vs. function words (section A4). This demonstrates that negative region differentiation is a systematic pattern across entangled neurons.
>
> We have revised the draft to make these contributions more apparent and flow more naturally. Taken together, the revised paper now more clearly shows that negative pre-activations are an active computational substrate for syntax, establishing a mechanistic and sign-dependent effect distinct from previous grammatical studies and prior work on Wasserstein neurons.

---

> > ### Author Response · Authors · 2025-11-21
> > **Response to Reviewer LJm9 (3/4)**
> >
> > ### 2. Grammar attribution via non-grammar benchmarks
> >
> > We agree that establishing grammar attribution is crucial to our work, and appreciate the reviewer’s focus on this point.
> >
> > First, to address the comment that grammar could simply be the first thing that is lost when a model degenerates, we directly control for this using our perplexity matched control in the original submission. As a summary, we observed that ablating only the negative pre-activations in 1% of the top WD neurons in each layer caused significant perplexity increase. We then found that ablating a much larger proportion of low WD neurons in the same way in each layer (40% in Llama, 50% in Mistral, and 20% in Qwen) yielded the same perplexity increase. We then compare model performance on the grammatical benchmarks BLiMP and TSE under each ablation. Despite both ablations causing the same perplexity increase, only in the top 1% Wasserstein neuron ablation is there an associated degradation in grammatical performance (Figure 3c). Thus, grammar is not always the first capability that is lost when a model degenerates to the same amount, as two equally damaging perturbations can have dramatically different grammatical effects. We have also revised our description of this process to remove confusion (L213).
> >
> > Second, we agree with your point on non-grammatical benchmarks being an important part of the story that our original submission did not include. To address this, we have run an additional panel of eight benchmarks on Llama 3.1 8B with our top 1% WD neuron negative pre-activation ablation, perplexity-matched bottom 40% WD neuron ablation control, and random 1% neuron ablation controls. We use ARC Challenge, ARC Easy, BoolQ, HellaSwag, PIQA, SciQ, TruthfulQA, and WinoGrande to assess scientific reasoning, question answering, reading comprehension, and general reasoning. These benchmarks are all multiple choice to remove the confound of generation ability from performance. This is now Figure 4 in our text.
> >
> > We find that, compared to the perplexity matched control, ablation of the negative pre-activations of the top 1% WD neurons produces much less degradation in model performance on these benchmarks. On average, error increased by 4% for the Wasserstein neuron ablation, whereas it increased by 11% for the perplexity-matched ablation. In seven out of eight benchmarks, the perplexity-matched case exhibited worse error. This stands in stark contrast to the performance on grammatical benchmarks, where the 1% WD neuron ablation yields significantly worse performance than the perplexity-matched control. Through this, we demonstrate a double dissociation phenomenon that strengthens our causal claim of the negative pre-activations of Wasserstein neurons being significantly implicated in syntax.
> >
> > We thank the reviewer for encouraging this additional evaluation. The new results now provide a substantially clearer and stronger attribution.
> >
> > ## Response to minor remarks
> >
> > 1. We originally used the phrase "(a, b) share the same legend" to indicate that the colors in both (a) and (b) correspond to the same models (red for Llama, green for Mistral, blue for Qwen) in Figure 3. We have now updated the caption to say this explicitly to avoid confusion (L226).
> >
> > 2. We appreciate the feedback in terms of writing style, and have now significantly revised the text to improve clarity, organization, and flow.

---

> ### Author Response · Authors · 2025-11-21
> **Response to Reviewer LJm9 (4/4)**
>
> ## Response to questions:
>
> ### 1. Reframing the conclusion
> We appreciate the reviewer's concern that the original conclusion may have sounded derivative of the general statement "important neurons matter." We have clarified it to emphasize that our findings characterize how the negative pre-activation region itself functions, going well beyond that generic premise.
>
> Specifically, the conclusions we highlight in the revision are not logical consequences of "Wasserstein neurons are important." That premise would predict only that perturbing these neurons harms performance. It does not predict:
>
> 1. Sign-specificity: selectively removing negative pre-activations, which are generally considered inert, in entangled neurons uniquely harms grammar, while leaving non-grammatical tasks comparatively unaffected.
> 2. Double dissociation: where the perturbations of the 1% entangled group vs the perplexity matched control produce opposite effects on grammatical and non-grammatical benchmarks, which cannot be derived from neuron importance alone.
> 3. Mechanistic localization: grammatical impact originates in early MLP layers, composes across depth, and is tightly linked to the use of negative differentiation.
> 4. Within neuron computation: individual neurons perform negative differentiation, mapping similar inputs to different depths of negativity, rather than relying on more general "large activation" patterns.
> 5. Sign vs magnitude separation: flipping the sign of negative values, thereby preserving the magnitude, is more damaging than zeroing either the positive or negative sides.
>
> These conclusions are not simply "more ablations on the same neurons." Rather, they are distinct claims that the ablations, in combination with token-level, layerwise, and training dynamics analysis, allow us to draw. Our revised Conclusion now better highlights these mechanistic inferences. We hope this clarifies that the contribution of the paper is to identify and characterize a sign-specific computation in the negative pre-activation region, rather than simply to show that certain neurons are important.
>
> Finally, we would like to sincerely thank the reviewer. Your feedback has led to substantial revisions that now more clearly highlight our contributions beyond prior work. We hope the updated manuscript addresses your concerns.

---

> ### Comment · Reviewer_LJm9 · 2025-11-24
>
> I thank the authors for considering the criticism and clarifying the missing pieces of the analysis. The newly provided results address the question of whether a direct connection to grammar can be drawn, which led to my drastic increase in score.

---

> > ### Author Response · Authors · 2025-11-26
> > **Thank you!**
> >
> > Thank you very much for your appreciation of our new results and for raising your score! We are grateful for your initial feedback, which has directly lead to improvements in many aspects of the paper.

---

### Official Review · Reviewer_9wYS · 2025-10-30

**Soundness:** 3
**Presentation:** 3
**Contribution:** 3
**Rating:** 6
**Confidence:** 3

**Summary:**

The paper shows the importance of wasserstein neurons within LLMs for syntactic processing. Precisely, wasserstein neurons are shown to map similar input vectors to different negative output values in LLMs equipped with smooth activation functions. Such neurons emerge very early during retraining and are more frequent in larger models. Ablating wasserstein neurons results in degraded syntactic capabilities, as reflected by the BLIMP and TSE benchmark scores. Furthermore, wasserstein neurons are shown to affect most the LLM’s performance on ellipsis and subject agreements, most crucially at early layers. Lastly, negative differenntiation is shown to be a mechanism for syntactic processing, forming negative-negative pairs.

**Strengths:**

* The paper is creative and shows a large set of results.
* The paper links elegantly syntax and single neurons, which is a unique and well executed finding.
* The writing and the figures are clear and precise.

**Weaknesses:**

* The correlation between the pretraining emergence of Wasserstein neurons and TSE is not convincing alone, there could be a lot of possible confounds during pretraining.

* There is no link with the mechanistic understanding of syntactic processing. For example, understanding *how* these neurons combine to  parse syntactic trees.

* There is no automated method for knowing what syntactic features drive each neuron, this would allow to list all wasserstein neurons and their associated syntactic pattern.

* The proposed mechanism is specific to LLMs with smooth activation functions (GELU/SiLU), but, is this condition necessary to acquire syntax? However, ReLU-based LLMs such as OPT also exhibit clear syntactic competence, suggesting that smoothness in the negative regime is not necessary for syntax acquisition. This raises the possibility that negative differentiation is one implementation path rather than a universal requirement for syntactic encoding.

**Questions:**

* When computing WD with respect to the unit Gaussian, where is this gaussian centered? Does this not modify the WD?
* Figure 2d, what is “syntactical scaffolding experience”? Why are determiners, punctuation … more important for syntax?
* It is quite surprising that syntactic processing happens at early layers within the MLP blocks and not in middle layers in attention layers (as previously suggested by probing works)? Is there any explanation for this?

---

> ### Author Response · Authors · 2025-11-21
> **Response to Reviewer 9wYS (1/2)**
>
> Thank you very much for reviewing our work and for your thoughtful comments! We are glad that you found the results creative and the syntax–neuron link elegant. Below we address each of your concerns and clarify how the revised paper responds to them.
>
> ## Response to weaknesses
>
> ### 1. Correlation between TSE and training
>
> We agree that the original emergence vs TSE panel alone could only be interpreted as correlational (Figure 2d). In the revision, we now explicitly frame Figure 2d as a motivating result between Wasserstein neurons and grammatical competency over training (L191), and we do not treat it as causal evidence. Our Figure 7c, d (Figure 6c, d in original submission) are meant to establish the causality of this relationship,  and we now explicitly make this point in the text (L422). To summarize, across many Pythia 1.4B checkpoints over training, we clamp the negative pre-activations of a fixed cohort of Wasserstein neurons and measure the resulting BLiMP/TSE error. The error increase grows consistently with the cohort’s mean WD at that checkpoint, while random and perplexity-matched controls (plotted at the same WD for reference) show little to no correlation. This provides a more causal demonstration that as Wasserstein neurons acquire their characteristic non-Gaussian structure, the model increasingly relies on them for grammar.
>
> ### 2. Full mechanistic understanding of processing
>
> We acknowledge and agree that we do not yet have a full circuit-level account of how these neurons implement syntactic trees. Our goal in this paper is to rigorously establish that negative pre-activations in a sparse subset of entangled neurons are functionally used for syntax, which was previously unknown. To move toward mechanism, we present negative differentiation of NN pairs as a potential component of how Wasserstein neurons encode syntax (Figure 7, A6). We view recovering the full circuit (e.g., explicit tree-building pathways) as exciting future work rather than a necessary result of the current paper.
>
> ### 3. Automated feature detection
>
> We agree that a fully automated mapping from Wasserstein neurons to syntactic features would be extremely valuable, but we view it as slightly beyond the scope of this work, which focuses on the existence and functional importance of the negative pre-activation region. To bridge toward such a system, we extended the differentiated token pair analysis to many more Wasserstein neurons in Pythia 1.4B (Section A.4, Figure A5). For each neuron, we examined its top differentiated token pairs, computed POS frequencies for the constituent tokens, and compared these frequencies against their base rates in WikiText 2. We find consistent and interpretable enrichment patterns that differ across neurons: some neurons predominantly differentiate determiners and punctuation, others emphasize past-tense verbs, others separate adpositions from determiners, and others isolate numerals. These POS enrichment statistics suggest that many Wasserstein neurons implement coherent syntactic distinctions, and they provide a natural foundation for a fully automated taxonomy, which we now highlight explicitly as exciting future work (L1043).
>
> ### 4. Negative differentiation as one of many paths toward syntax
>
> We agree that smooth activations and negative differentiation cannot be necessary universally, since ReLU-based models like OPT also achieve strong syntactic competence. Our claim is instead that, in smooth-activation LLMs, negative pre-activations in Wasserstein neurons constitute an important and previously overlooked implementation pathway for syntax. We now stress this in the related work by framing our mechanism as complementary to prior accounts of locating syntax (L454).

---

> > ### Author Response · Authors · 2025-11-21
> > **Response to Reviewer 9wYS (2/2)**
> >
> > ## Response to questions:
> >
> > ### 1. Distribution normalization
> >
> > Yes, we normalize the neuronal output distribution to have mean zero and unit variance (L121) to account for differences in the position of the distribution. Doing so helps the WD metric focus exclusively on shape. We apologize if this was not clear.
> >
> > ### 2. Scaffolding definition
> >
> > We have clarified the text around Figure 3d to make this notion more precise. By “syntactic scaffolding tokens” we mean function-like items, such as determiners ("the", "a"), auxiliaries, particles, adpositions, and punctuation, that contribute primarily to the structural organization of the sentence rather than to its lexical content, which is why they are generally regarded as core operators that regulate syntactic relations. In our token-level analysis, excess surprisal under 1% Wasserstein neuron negative pre-activation ablation is concentrated on these POS and dependency classes, while content words (nouns, adjectives) are comparatively spared. We hope that this addresses your question.
> >
> > ### 3. Our results in the context of attention
> >
> > We share your surprise and agree that this is an interesting tension with probing work that often finds syntax in middle attention layers. Our experiments focus specifically on MLP gate/up neurons and do not attempt to localize syntax within attention heads, and we now clarify this in the related-work section (L452). It is possible that our findings identify an early MLP mechanism that provides syntactic scaffolding signals (via negative differentiation in Wasserstein neurons) that attention and later layers can possibly build upon, rather than contradicting the existence of syntactic structure in attention. Exploring how these early MLP features interact with attention-based representations is a natural and exciting next step.
> >
> > Thanks again for your positive view of our paper and for your useful feedback! We are particularly encouraged by how interpretable the differentiated tokens seem to be based on part of speech, providing exciting future work. We hope to have addressed your concerns through our results and the updated paper.

---

### Official Review · Reviewer_abiD · 2025-10-31

**Soundness:** 3
**Presentation:** 3
**Contribution:** 3
**Rating:** 6
**Confidence:** 4

**Summary:**

The authors study "Wasserstein" neurons, which exhibit strongly non-Gaussian distributions of preactivations. They show that for models with non-ReLU activation functions, negative preactivations are functionally important through sign-specific ablations, developmental analysis across checkpoints, and other experiments. The authors provide additional evidence for these neurons being especially implicated in predicting grammatical techniques by targeting their analysis to grammatical benchmarks.

**Strengths:**

- **Well-motivated**: The introduction is clearly written. The discussion on the need to reexamine the distribution of negative pre-activations for non-ReLU is strong.
- **Diverse set of models**: I appreciate that the authors have included a diversity of LLMs.
- **Self-explanatory figures**: The figure captions are clearly written and self-explanatory, making this easy to follow.

Full disclosure: I reviewed an earlier workshop version. From memory, this revision is a substantial improvement in cohesion.

**Weaknesses:**

A. WD may act as a proxy for skew/kurtosis rather than "entanglement."

In many places, the authors treat "high WD from a Gaussian baseline" ("WD score") as equivalent to "highly multimodal" or "strongly entangling." This is implicitly justified via the observation that WD and mapping difficulty (MD)  are highly correlated: i.e., neurons with high WD score are likely to map similar input vectors to widely separated preactivations. However, "high WD score" is also compatible with "non-negligible higher moments." This is not discussed, which is a source of confusion and leads to implicitly overstating the strength of the Wasserstein neurons as "entangled neurons" hypothesis.

Minimum fix: Some additional comment clarifying the alternative hypothesis (high skew/kurtosis/higher moments) would be enough to resolve this concern. The authors could mention as a limitation that the methodology cannot pick up the distinction between heavy tails and multimodality and that this weakens the "entanglement" interpretation.

Maximum fix: Run an additional comparison between WD and skewness/kurtosis.

Point of clarification: Section 2.1 defines MD by input‑difference / output‑difference averages. If that is indeed the implemented metric, high MD means _low_ output separation for similar inputs. This is a bit confusing because the text uses (lines 126–127) "MD of a neuron to summarize how far apart neurons map similar inputs."

B. The argument for WNs affecting grammar over other kinds of structure is weak(ly presented).

The grammar hypothesis is introduced in section 2 by demonstrating that WD correlates with TSE (a specific grammatical benchmark) performance over training. In this section, the authors never discuss any alternative explanation (e.g., WNs are helpful for general-purpose natural language processing). Towards the end of the section the authors ask (lines 195–198): "is [this] structure merely a byproduct, or is it causally necessary for grammatical behavior?" This is a sudden and unexpected jump.

Evidence for the grammar hypothesis is presented in section 3 and Fig 3c & d and section 3. I find this mostly convincing, though I would like to see an additional comparison like Figure 3c against a non-grammatical control benchmark (from my current understanding, the authors have not yet ruled out the possibility of WNs being important for grammar *as well as* being important to other roles – though figure 4d does point in this direction).

Line 275–278 "Cumulative ablations raise error across all syntactic classes, consistent with a role for Wasserstein neurons as more general-purpose carriers of grammatical information rather than specifically detecting individual phenomena" There is no discussion of any control assessing the effects across non-syntactic categories. Perhaps this evidence is implicitly presented in figure 4, but it is not clear to me where to look for this. I would appreciate if the authors can comment on this.

Minimum fix: The paper would be much easier to understand if the grammar hypothesis is introduced more naturally. Another option would be to simply foreshadow the grammar hypothesis in section 2 and promise that it will be justified later.

Maximum fix: The paper would benefit from repeating 4c with additional non-grammatical benchmarks, and providing further detailed examples illustrating the effects of ablations on individual samples in Fig 4d. This could be limited to an appendix.

Small additional note: when comparing to perplexity‑matched controls, please state clearly whether the layer distribution and number of perturbed units per layer are matched. If not, a sentence explaining that the current matching is still incomplete would pre‑empt reader confusion.

C. Treatment of negative-negative separation is relatively weak.

There's an obvious confounder in the analysis of NN vs PN vs PP differentiation. Zero-clamping negative preactivations collapses the distinction between two samples that both have negative preactivations that are only distinguished in magnitude. The same is not true for two samples that differ in preactivation sign or where both inputs have negative preactivations. As such, it is very unclear to me whether the observed higher correlations between NN ablations and damaged performance tell us anything about whether NN-differentiated samples are more important for grammar. The correct baseline would also involve ablating the particular positive activations for the PN or PP pairs.

Minimum fix: Mention the confounder explicitly and point to it as a limitation of the current design.

Maximum fix: Run additional experiments. For example, you could run (i) a matching positive‑clamp for the PN/PP baselines in the same layers, or (ii) sign‑flip/abs interventions to test whether sign vs. magnitude drives the effect.

Suggestion: I find the NN/PN/PP differentiation analysis to be relatively weaker than the rest of the text. I would recommend moving section 7 (and 6?) to the appendix (along with a clearer mention of confounders), and using the remaining space to strengthen the case for the preceding sections (esp. the grammar hypothesis).

If the authors implement all of my minimum fixes and one or two of the maximum fixes, I will raise my score to an accept. I think including all of the maximum fixes is unlikely to further raise my score to a strong accept, but I would still recommend the authors consider these additional changes.

**Questions:**

- Besides the fact that WD is more neuron-focused than MD (lines 127–129), is the main reason to favor WD over MD that WD is more computationally efficient? It seems like it might otherwise be better to compute aggregate MD scores for each neuron since this would get closer to the question of entanglement.

Minor improvements:

- (This is related to weakness A) The definition of Wasserstein neurons and "non-Gaussian" structure could be improved in Sec 2.1 and Fig 1. Phrasing such as (lines 76–78) "the non-Gaussian structure of Wasserstein neurons appears in the positive region ... [for GELU] the non-Gaussian structure is concentrated in the negative pre-activation region"  is confusing because the distribution as a whole is clearly non-Gaussian. I believe the authors mean to say something like "the bulk of preactivations occupies a large Gaussian peak. The mode of this peak is concentrated in the negative/positive region respectively for ReLU and non-ReLU networks." It would be clearer if the authors discussed this purely in terms of the largest mode of the distribution.
	- Line 63–66 "the deviation from Gaussianity concentrates in the negative region of the pre-activation space ... ReLU-based models ... do not exhibit comparable structure in the negative space" is clearly wrong as stated since the ReLU-based neurons also have clear non-Gaussian structure across the positive and negative range of activations.
- Lines 55–57 "Nevertheless, many interpretability workflows retain the ReLU-era heuristic, equating activity with positive pre-activations and implicitly treating the negative region as inert." I question whether this is true. I believe many workflows like SAEs are sensitive to negative activations. Maybe the authors believe that the field still implicit acts with this expectation (which would require weakening the language) or they can provide some references to examples of real-world workflows making this assumption.
- Lines 16–17: "We show that this negative region is functional rather than simply favorable for optimization." What does this mean?
- Line 19 "entangled neurons" is not defined until line 38
- Line 161: "mildly more expressive in their distributions" what does this mean?

---

> ### Author Response · Authors · 2025-11-21
> **Response to Reviewer abiD (1/2)**
>
> We would like to sincerely thank the reviewer for their very thorough assessment of our work! We appreciate your positive view of our motivation and presentation. Based on your helpful comments, we have run additional experiments to address your feedback in an attempt to conduct the maximum fix for each point, and have updated our submission to incorporate these changes. Below, we respond to each point that you raise.
>
> ## Response to weaknesses:
>
> ### A. WD vs skew vs kurtosis
>
> We agree that WD can be confounded by the skew or kurtosis of the distribution. To further investigate this, we calculate the WD, MD, absolute skew (asymmetry), and excess kurtosis for every neuron in each gate projection layer of Llama 3.1 8B (Section A.1).
>
> We find that in all layers, WD and MD closely track one another (Figure A1). Because our analysis utilizes the most entangled neurons, we then calculate the Jaccard index between the top 1% of neurons ranked by each metric and measure agreement, with a Jaccard index of 0 indicating no agreement and 1 indicating complete agreement. Across all layers, the agreement of the top 1% of neurons as calculated by WD and MD are the highest, compared to WD and asymmetry, WD and kurtosis, MD and asymmetry, and MD and kurtosis (Table A1). In the text, we have now added that the concern that skew and kurtosis could be potential confounds was further investigated (L133).
>
> Finally, we repeated our main causal experiments in Figure 3 on Llama 3.1 8B, but using MD directly to select the most or least entangled neurons instead of the WD as a proxy. All qualitative results are identical, showing that WD is indeed a suitable proxy for MD (Figure A3).
>
> Also, thank you very much for catching our error for the MD definition. We indeed meant to define MD as normalized outputs divided by normalized inputs, but mistakenly reversed the definition in the text. All results used the correct implementation of MD. This typo has now been amended (L129).
>
> ### B. Presentation on evidence for grammar, and non-grammar benchmarks
>
> Thank you for your feedback on the flow of the topics. To better introduce our concepts, we have changed the wording in the section that introduces assessing TSE during training to more directly ask the question we are interested in regarding grammar (L190).
>
> Additionally, to validate the specificity of syntax in the negative pre-activations of Wasserstein neurons, based on your suggestion, we have run an additional panel of eight benchmarks on Llama 3.1 8B with our top 1% WD neuron negative pre-activation ablation, perplexity-matched bottom 40% WD neuron ablation control, and random 1% neuron ablation controls. We use ARC Challenge, ARC Easy, BoolQ, HellaSwag, PIQA, SciQ, TruthfulQA, and WinoGrande to assess scientific reasoning, question answering, reading comprehension, and general reasoning. These benchmarks are all multiple choice to remove the confound of generation ability from performance. This is now Figure 4 in our text.
>
> We find that, compared to the perplexity matched control, ablation of the negative pre-activations of the top 1% WD neurons produces much less degradation in model performance on these benchmarks. On average, error increased by 4% for the Wasserstein neuron ablation, whereas it increased by 11% for the perplexity-matched ablation. In seven out of eight benchmarks, the perplexity-matched case exhibited worse error. This is in stark contrast to the performance on grammatical benchmarks, where the 1% WD neuron ablation yields significantly worse performance than the perplexity-matched control. Through this, we demonstrate a double dissociation phenomenon that strengthens our causal claim of the negative pre-activations of Wasserstein neurons being significantly implicated in syntax.
>
> Regarding L275, we meant that, unlike in BLiMP in which certain categories did not exhibit a large degradation in performance, all TSE categories did. We realize this was unclearly presented, so we have now reworded this sentence (L330).
>
> Regarding the comparison to perplexity matched controls, we apologize if our definition was confusing previously. We have now amended the section to better clarify the exact procedure (L213). Briefly, what we did is, in each gate projection layer, select the lowest m% of neurons based on WD to ablate the negative pre-activations, where m is the same value across all layers. Then, m was gradually increased, until the perplexity degradation was matched to that of the 1% Wasserstein ablation.

---

> > ### Author Response · Authors · 2025-11-21
> > **Response to Reviewer abiD (2/2)**
> >
> > ### C. Negative differentiation concerns
> >
> > We agree that negative-only pre-activation ablations disproportionately collapse NN pairs, since these pairs differ only in negative magnitude, and this introduces a structural confound in our original NN/PN/PP analysis. We now make this limitation explicit in the text (L1169).
> >
> > To partially mitigate this confound and to probe whether sign versus magnitude drives the observed grammatical vulnerability, we conducted three additional ablations beyond zeroing negative pre-activations (Section A.3). In Llama-3.1-8B, we tested:
> >
> > 1. replacing the positive branch of SiLU with ReLU
> > 2. zeroing positive pre-activations
> > 3. flipping the sign of negative post-activations while preserving their magnitude
> >
> > These controls serve distinct purposes. The ReLU-positive ablation tests whether curvature in the positive branch of SiLU contributes anything meaningful, the positive-zeroing ablation assesses the relative importance of the positive and negative regions, and the sign-flip ablation isolates the contribution of sign itself, while keeping magnitudes approximately constant.
> >
> > The results sharply differentiate these cases. Replacing SiLU’s positive curvature with ReLU yields almost no degradation, as expected. Zeroing the positive signal produces a larger degradation than zeroing the negative signal, consistent with the known functionality of positive branches carrying the bulk of a model's overall information. Most strikingly, flipping the sign of negative post-activations causes extreme damage: flipping the negative outputs of only the top 2% WD neurons increases perplexity from 6.5 to 2200, exceeding even the damage from zeroing positive activations at the same ablation rate. Because this intervention preserves magnitude, it shows that the sign of negative responses in Wasserstein neurons carries essential functional information, and that inverting that sign is more destructive than removing the signal entirely.
> >
> > Nonetheless, given the potential confound and for the purpose of streamlining the main narrative, we have moved Section 7 to the appendix (Section A.5) and replaced it in the main text with the new non-grammatical benchmark results (Figure 4). Together, these changes make clear that the NN/PN/PP analysis is more suggestive rather than fully causal, and that the main claims rely instead on sign-specific ablations and task-level double dissociation results.
> >
> > ## Response to questions
> >
> > ### 1. WD or MD
> >
> > You are correct that the main reason we previously favored WD over MD is that it was more computationally efficient to run. However, we have now made our calculation of MD much more efficient, and so in future work we will pivot to that metric primarily, since we agree that this metric more directly assesses entanglement. To validate our current workflow using WD, as mentioned previously, we obtained the same qualitative results during causal ablations when selecting neurons based on MD as we did when selecting based on WD (Section A.1.3).
> >
> > ## Response to minor improvements
> >
> > 1. Thank you for mentioning this, the idea you presented is indeed what we were trying to get across. Based on your wording, we have now revised the phrasing for clarity (L77).
> > 2. Thank you for this note, we have softened our language (L67). We originally mean that, broadly speaking, the negative pre-activation region for the OPT neuron was slightly less multimodal, though we realize it is not fully Gaussian.
> > 3. We appreciate your clarifying question on current interpretability workflows. We originally meant that many neuron based workflows consider the inputs that most positively activate a neuron as what said neuron attends to (AlKhamissi et al 2024, Gurnee et al 2023). For SAEs, most works seem to be looking at the output of the MLP block or residual layers, rather than preceding or following the activation function within an MLP block (Rajamanoharan et al 2024, Lieberum et al 2024, Gao et al 2024). We have now better clarified our language here as well (L40).
> > 4. For this line, we meant to emphasize that pre-activations have a specific mechanistic function. This is opposed to their only being useful for training optimization. We have now clarified it in the text (L18).
> > 5. We have now re-ordered the introduction to naturally talk about “entangled neurons” more naturally, after the idea of negative pre-activations are introduced.
> > 6. This was meant to indicate that for larger models, neurons on average tended to have a greater WD, even those that are not Wasserstein neurons. This has been clarified with more precise statistical language (L163).
> >
> > Thank you again for your time in reviewing our paper and your very useful and extensive feedback. We feel that, as a result of the changes we made to the paper based on your suggestions, the paper has been substantially improved and its core contribution strengthened. We hope that the responses and associated experiments addressed your concerns.

---

### Official Review · Reviewer_6mmb · 2025-11-01

**Soundness:** 3
**Presentation:** 4
**Contribution:** 3
**Rating:** 8
**Confidence:** 4

**Summary:**

This paper argues that a tiny, entangled subpopulation of MLP "Wasserstein neurons" in LLMs computes in the negative pre-activation region of smooth activations, and that this sign-specific behavior is mechanistically important for syntax. Evidence: (1) WD/"mapping difficulty" characterization across models; (2) sign-specific ablations that zero only negative pre-activations for the top-WD neurons cause large PPL increases and large drops on BLiMP/TSE, unlike random or perplexity-matched controls; (3) effects localize to functional POS tokens and early layers, compounding across depth; (4) training-dynamics: Wasserstein neurons emerge early and the same sign-specific ablation harms grammar more as training proceeds. ReLU models (OPT) lack the negative-region structure, consistent with clamping.

**Strengths:**

1. Clear, testable mechanistic claim (negative pre-activation computation) linked to syntax; the sign-specific intervention is surgical and well-motivated for non-ReLU models.

2. The paper also provides compelling causal evidence. That is, clamping negatives of top-WD neurons yields large BLiMP/TSE drops even when perplexity is matched by ablating many low-WD neurons; random controls are much smaller.

3. The paper demonstrates cross-model coverage (Pythia, Llama-3.1-8B, Mistral-7B-v0.3, Qwen-3-8B) and ReLU vs non-ReLU comparison.

**Weaknesses:**

1. Zeroing negatives changes the neuron’s transfer function (effectively ReLU-izing selected channels), which can alter residual/RMSNorm statistics and induce broad PPL rises unrelated to "syntax-specific" computation. My worry is that the perplexity-matched control (many low-WD channels clamped) helps, but it does not fully control for distributional/variance changes in the same high-WD channels.

2. The figure-level observation that OPT (ReLU) lacks negative-region structure is interesting; strengthen by running the same sign-specific ablation (no-op in ReLU) plus a mirror positive-only clamp in non-ReLU models to show asymmetry is truly about negative region utility rather than general clipping.

3. The negative-negative (NN) separation result is striking; however, the pipeline samples top MD pairs among similar inputs and counts sign patterns. To exclude lexical/position confounds, you can do better control for token frequency, position, and subword overlap.

4. Grammar drops are strong; it would be great if you could also report non-syntactic tasks (e.g., TruthfulQA/QA/calibration) to establish specificity: are you removing general capacity or truly syntax-critical computation? The POS analysis suggests scaffolding specificity, but a broader behavioral panel would solidify the claim.

**Questions:**

1. In training dynamics, if you freeze the early high-WD cohort vs. others, do grammar curves diverge as predicted?

2. Could you report CIs/seeds for BLiMP/TSE?

---

> ### Author Response · Authors · 2025-11-21
> **Response to Reviewer 6mmb (1/2)**
>
> Thank you for your positive assessment of our work! We appreciate that you found our causal evidence of the functionality of negative pre-activations compelling. Based on your feedback, we have run additional experiments to strengthen our claims and have updated our submission accordingly. We detail our responses to your comments, as well as the changes we have made in response to each, below.
>
> ## Response to weaknesses
>
> ### 1. RMSNorm statistical confounds
>
> We agree that zeroing negative values does change the statistics that feed into normalization, which could induce unintended effects. However, we believe that, because we only clamp the negative values of approximately 1% of all neurons in our experimental ablation, the effect on normalization statistics should be minor. As you point out as well, the perplexity-matched control clamps a much greater number of neurons, yet has little to no effect on grammar benchmarks. To further investigate your question, we have now conducted additional experiments with different sign-specific activation function ablations, which we describe in the next point.
>
> ### 2. Additional sign-specific ablations
>
> First, we confirm that zeroing negative pre-activations in a ReLU based model causes no change, as expected. In Llama 3.1 8B, we then test three additional ablations to the activation function beyond zeroing negative pre-activations, specifically:
>
> 1. setting positive pre-activations to use the ReLU activation function rather than SiLU
> 2. zeroing positive pre-activations
> 3. flipping the sign of negative post-activations
>
> We use the first additional ablation to check whether something about the curvature in the positive region of the SiLU activation function. We use the second to examine how significant the positive side is compared to the negative side. Finally, we use the last ablation to both check whether the sign of the negative value matters in addition to its magnitude, and to help control for potential normalization issues, as the magnitude is unchanged, and so the normalization statistics should not change either.
>
> As an initial confirmation, the curvature of the SiLU in the positive region is not particularly salient, as replacing that with ReLU yielded very little change to model performance. By contrast, zeroing the positive signal of Wasserstein neurons is substantially more damaging than applying the same intervention to random neurons and also more damaging than zeroing the negative signal, consistent with the known functionality of positive branches carrying the bulk of a model's overall information.
>
> Strikingly, reversing the sign of negative post-activations yields substantial model damage, moreso than even zeroing out positive activations at 2% ablation. Just by reversing the negative signs of the top 2% WD neurons, perplexity climbs from 6.5 to 2200. Crucially, this intervention preserves the magnitude of the negative responses and thereby largely preserves their contribution to normalization statistics, while inverting only the sign. Together, these controls show that the sign of negative activations in Wasserstein neurons carries essential information, and inverting it is more harmful than removing it altogether (Section A.3).
>
> ### 3. Lexical confounds in the MD analysis pipeline
>
> Thank you for raising the possibility that NN separation patterns could be influenced by lexical frequency, positional structure, or subword overlap in natural text. Because our MD-based sampling uses real model inputs, fully controlling for these factors is indeed difficult without introducing artificial data distributions.
>
> To address this concern, we conducted an additional analysis across a broader set of Wasserstein neurons (Section A.4). For each neuron, we examined the top differentiated token pairs and compared the POS frequencies of these tokens against their base rates in the corpus. If NN (or PN/PP) structure were driven primarily by lexical or positional confounds, we would expect the differentiated tokens to resemble the global corpus distribution.
>
> Instead, we find systematic and interpretable POS enrichment patterns that differ across neurons: some neurons predominantly differentiate determiners and punctuation, others emphasize past-tense verbs, others separate adpositions from determiners, etc.
>
> These enrichments are not explained by global token frequency or positional bias, and they vary meaningfully across neurons. This indicates that the differentiated-pair structure reflects neuron-specific syntactic specialization, rather than artifacts of lexical frequency or position.
>
> We have updated the text to present these enrichment results and to note that they substantially reduce the likelihood that NN patterns arise from surface-level confounds.

---

> ### Author Response · Authors · 2025-11-21
> **Response to Reviewer 6mmb (2/2)**
>
> ### 4. Non-grammar benchmarks
>
> We agree with your comment that determining how the performance of non-grammatical benchmarks is impacted by our ablations. Based on your suggestion, we have run an additional panel of eight benchmarks on Llama 3.1 8B with our top 1% WD neuron negative pre-activation ablation, perplexity-matched bottom 40% WD neuron ablation control, and random 1% neuron ablation controls. We use ARC Challenge, ARC Easy, BoolQ, HellaSwag, PIQA, SciQ, TruthfulQA, and WinoGrande to assess scientific reasoning, question answering, reading comprehension, and general reasoning. These benchmarks are all multiple choice to remove the confound of generation ability from performance. This is now Figure 4 in our text.
>
> We find that, compared to the perplexity matched control, ablation of the negative pre-activations of the top 1% WD neurons produces much less degradation in model performance on these benchmarks. On average, error increased by 4% for the Wasserstein neuron ablation, whereas it increased by 11% for the perplexity-matched ablation. In seven out of eight benchmarks, the perplexity-matched case exhibited worse error. This is in stark contrast to the performance on grammatical benchmarks, where the 1% WD neuron ablation yields significantly worse performance than the perplexity-matched control. Through this, we demonstrate a double dissociation phenomenon that strengthens our causal claim of the negative pre-activations of Wasserstein neurons being significantly implicated in syntax.
>
> ## Questions
>
> ### Training dynamics
>
> Freezing the early high-WD cohort during training would indeed provide a clean causal test, but it requires retraining from scratch with a modified forward pass and is unfortunately beyond our current compute budget.
>
> However, we have performed the closest feasible analysis: for various checkpoints during Pythia 1.4B training, we apply the same negative pre-activation ablation to the high-WD cohort and track the resulting BLiMP/TSE performance (Figure 6c, d). As training progresses, the grammar damage induced by this fixed ablation increases, and the gap between Wasserstein-neuron ablation and random and perplexity matched control ablations widens.
>
> This shows that as the model learns, these high-WD neurons become increasingly specialized for grammatical structure, which is consistent with what a freezing experiment would test. We consider a full freezing-based causal test as exciting future work.
>
> ### Benchmark CIs
>
> We now provide the raw benchmark scores with their standard errors in Section A.2. We ran our benchmarks at temperature = 0.
>
> Thank you again for your support of our work! We appreciate your feedback and suggestions, and we believe that our work's core claims have now been further strengthened based on them.

---

### Comment · Area_Chair_rjgm · 2025-11-25
**Please discuss**

Several reviewers have not responded to the authors' rebuttals. Please read and respond to them. Have the rebuttals addressed your concerns or clarified anything?

---

### Author Response · Authors · 2025-12-01
**Thank You and Rebuttal Summary for the New Area Chair**

Dear AC and Reviewers,

We greatly appreciate your overseeing of the rebuttal process and your significant engagement with the work. Here, we briefly outline the additional experiments we conducted during the rebuttal process and highlight how they strengthen the core contributions while addressing the reviewers' concerns.

1. **Non-grammatical benchmarks as controls.** In addition to our original benchmarks on TSE and BLiMP, we now run our experimental setup on a panel of eight non-grammatical tasks. We observe that ablating the negative pre-activations of Wasserstein neurons causes comparatively less performance loss on these benchmarks than the same ablation in the perplexity matched control. Together, this exhibits a clear double dissociation phenomenon between syntactic and non-syntactic capabilities, strengthening our causal claim that the negative pre-activations of Wasserstein neurons are causally implicated in syntax as opposed to general, non-grammatical function (Figure 4). This addresses weakness 4 for Reviewer 6mmb, weakness B for Reviewer abiD (maximum fix), and weakness 2 for Reviewer LJm9.

2. **More sign-specific ablations.** In addition to our original ablation of negative pre-activations, we have also tested ablating positive pre-activations, replacing the positive side of SiLU with ReLU, and reversing the sign of the negative post-activation. While the replacement of the positive side of SiLU with ReLU has, as expected, very little impact on model performance, we have found that reversing the sign of negative post-activations to be positive actually yields substantial model damage, often more so than even zeroing out positive activations at the same ablation rate. This supports the notion that the sign of the negative activations is crucial information, and inverting it is more harmful than removing it altogether (Figure A4). This addresses weaknesses 1 and 2 for Reviewer 6mmb and weakness C for Reviewer abiD (maximum fix).

3. **Reordering of paper sections and other writing improvements to better frame contributions.** We reorganize the introduction to better highlight our work as an investigation into negative pre-activations, using Wasserstein neurons as a tractable subset of neurons to study rather than as the central object of study. We also re-frame the conclusion to focus on the complexity of the negative pre-activation space as well. Additionally, we better specify that Figure 2d is meant as a motivating result that is correlational rather than causal, and link Figure 7c and d directly back to Figure 2d as the causal evidence. This addresses weakness 1 of Reviewer 9wYS and weakness 1 of Reviewer LJm9.

4. **Further characterization of Wasserstein neurons.** In addition to the neuron in Figure 6, we also characterize the most differentiated token pairs of other Wasserstein neurons. We find many interpretable patterns, such as Wasserstein neurons that differentiate the token "and" from other tokens and those that attend to numeric tokens. The parts of speech that correspond to these tokens are substantially enriched when compared to their presence in the Wikitext 2 input text as a whole (Figure A5). This addresses weakness 3 of Reviewer 6mmb and weakness 3 of Reviewer 9wYS.

5. **Confounds of skew and kurtosis.** To ensure that we better account for the potential confounds that skew and kurtosis may have on Wasserstein distance (WD), we compared each neuron’s WD, mapping difficulty (MD), asymmetry (absolute skew), and kurtosis across all gate projections of Llama 3.1 8B. We find that WD and MD closely track one another across all layers (Figure A1). Furthermore, we find that the Jaccard index of the top 1% of neurons as ranked by WD and by MD are higher than the Jaccard index of the top 1% of neurons as ranked by WD and asymmetry, by WD and kurtosis, by MD and asymmetry, and by MD and kurtosis (Table A1). Finally, we repeat all experiments in Figure 3 using MD instead of WD as a selection metric, and observe the same qualitative results (Figure A3), better motivating the use of WD as a practical proxy for MD and for selecting non-Gaussian, entangled neurons. This addresses weakness A of Reviewer abiD (maximum fix).

In light of these changes, Reviewer LJm9 felt that "the newly provided results address the question of whether a direct connection to grammar can be drawn", leading to their "drastic increase in score" to a 6. Additionally, Reviewer abiD specified that "if the authors implement all of [their] minimum fixes and one or two of the maximum fixes, [they] will raise [their] score", and our revision implements the maximum fix for each of their weaknesses.

Overall, despite the unfortunate circumstances that prevented further discussion during the rebuttal phase, we appreciate the opportunity to have significantly improved our submission as a result of reviewer interactions. We once again thank the reviewers and the AC for their engagement with and consideration of our work.

---

### Meta-Review · Area_Chair_nyJD · 2026-01-05

**Summary:**

Reviewers generally found the paper’s central claim compelling: in smooth-activation LLMs, negative pre-activations in a small set of “Wasserstein neurons” play a causal role in syntactic performance, supported by targeted ablations and controls. The main concerns shaping the decision were (i) whether the effects are truly syntax-specific rather than general degradation (including normalization/statistical confounds), (ii) whether “Wasserstein distance” is meaningfully tied to entanglement vs higher-moment proxies, and (iii) novelty/framing relative to prior work on Wasserstein neurons.

**Reviewer Concerns:**

Addressed by the rebuttal / revision

Syntax-specificity vs general damage: Authors added an 8-task non-grammar benchmark panel and report a double dissociation (non-grammar tasks degrade more under perplexity-matched control than under Wasserstein-neuron negative ablation), directly targeting the key “general degradation” critique.

Normalization/statistical confounds & sign-specificity controls: Beyond negative clamping, the authors ran additional sign-specific interventions to argue that the sign of negative activations carries functional information and to mitigate concerns about RMSNorm-statistics.

WD vs skew/kurtosis proxy: Authors explicitly compared WD, mapping difficulty, skewness, and kurtosis, and replicated key results using MD to support WD as a practical proxy rather than an artifact of higher moments.

Framing/correlation vs causality: They clarified that the emergence-vs-grammar plot is correlational and later repositioned the evidence as causal support, improving interpretive hygiene.

Still outstanding / partially outstanding

Mechanistic depth remains limited: One reviewer’s request for a clearer “how” (e.g., links to syntactic tree parsing; automated identification of syntactic features per neuron) is only partially addressed via additional qualitative neuron characterizations and POS enrichment, but not a full mechanistic account.

Generalization of the proposed mechanism: The paper still reads as identifying a critical implementation path (especially for smooth activations) rather than establishing the necessity for syntax across architectures; this limitation is acknowledged but not fully resolved empirically.

Some “clean causal” training interventions remain infeasible; e.g., freezing-based tests were noted as computationally out of scope.

**Reviewer Scores:**

Reviewer 6mmb: remain 8.

Reviewer abiD: remain 6.

Reviewer 9wYS: remain 6.

Reviewer LJm9: from 2 to 6, explicitly reported a “drastic increase in score” after rebuttal results addressing grammar specificity.

---

### Decision · Program_Chairs · 2026-01-26

Accept (Poster)